# Global phosphoproteomic analysis reveals ARMC10 as an AMPK substrate that regulates mitochondrial dynamics

Zhen Chen[1], Caoqi Lei[2], Chao Wang[1], Nan Li[1], Mrinal Srivastava[1], Mengfan Tang[1], Huimin Zhang[1], Jong Min Choi[3], Sung Yun Jung [3], Jun Qin[3] & Junjie Chen[1]

AMP-activated protein kinase (AMPK) is a key regulator of cellular energy homeostasis. Although AMPK has been studied extensively in cellular processes, understanding of its substrates and downstream functional network, and their contributions to cell fate and disease development, remains incomplete. To elucidate the AMPK-dependent signaling pathways, we performed global quantitative phosphoproteomic analysis using wild-type and AMPKα1/α2-double knockout cells and discovered 160 AMPK-dependent phosphorylation sites. Further analysis using an AMPK consensus phosphorylation motif indicated that 32 of these sites are likely direct AMPK phosphorylation sites. We validated one uncharacterized protein, ARMC10, and demonstrated that the S45 site of ARMC10 can be phosphorylated by AMPK both in vitro and in vivo. Moreover, ARMC10 overexpression was sufficient to promote mitochondrial fission, whereas ARMC10 knockout prevented AMPK-mediated mitochondrial fission. These results demonstrate that ARMC10 is an effector of AMPK that participates in dynamic regulation of mitochondrial fission and fusion.

[1] Department of Experimental Radiation Oncology, The University of Texas MD Anderson Cancer Center, Houston, TX 77030, USA. [2] Hubei Key Laboratory of Cell Homeostasis, College of Life Sciences, Wuhan University, 430072 Wuhan, China. [3] Department of Molecular and Cellular Biology, Baylor College of Medicine, Houston, TX 77030, USA. These authors contributed equally: Zhen Chen, Caoqi Lei. Correspondence and requests for materials should be addressed to J.C. (email: jchen8@mdanderson.org)

AMP-activated protein kinase (AMPK) is a kinase complex that acts as a central regulator of cellular energy homeostasis in eukaryotes. It monitors ATP levels in cells. When the ratios of AMP:ATP and ADP:ATP increase, AMPK is activated and controls the activities of enzymes in a variety of pathways to ensure energy homeostasis. It switches on the glucose uptake and other catabolic pathways to generate ATP, while switching off the anabolic pathways to prevent the consumption of ATP, such as the conversion of glucose to glycogen[1]. AMPK also phosphorylates 3-hydroxy-3-methyl-glutaryl–coenzyme A reductase and glycerol-3-phosphate acyltransferase to block the synthesis of sterols and triglycerides, respectively[2]. These regulatory actions by AMPK ensure increased cellular ATP supplies and decreased ATP consumption. AMPK also modifies the mammalian target of rapamycin complex, which functions as the master switch in controlling cell proliferation and fate by inhibiting autophagy and apoptosis[3,4].

As a key regulator of many cellular processes, AMPK plays a central role in a variety of human diseases. Studies of AMPK in cancer, diabetes, and other human diseases verified its important roles in disease development[5–7]. Moreover, several compounds that have become therapeutic centerpieces seem to produce their protective and therapeutic effects by modulating AMPK signaling. For example, investigators are testing metformin and other agents that activate AMPK in the clinic as potential anticancer agents[7,8].

Discovery of AMPK substrates is critical for understanding AMPK functions and its applications in disease treatment. Several groups have used different strategies to identify AMPK substrates. For example, Shaw and colleagues, using 14-3-3 binding and AMPK substrate motif searching, identified several important AMPK substrates, such as ULK1, Raptor, and mitochondrial fission factor (MFF)[9–11]. Also, Brunet and colleagues combined a chemical genetic screen and peptide capture technique to identify AMPK phosphorylation sites[12]. James and colleagues reported on their global phosphoproteomic analysis of acute exercise signaling in human skeletal muscle and performed additional targeted AMPK assays and bioinformatics analysis to predict AMPK substrates[13]. Furthermore, Sakamoto and colleagues used an anti-AMPK motif antibody to discover AMPK targets[14]. Although these experimental approaches identified many AMPK substrates, defining the AMPK-dependent signaling network remains challenging because of the high background or noise level. Bioinformatics analysis is one way to filter data and uncover bona fide AMPK substrates. In this study, we reduced background by using AMPKα1/α2-double knockout (DKO) cells as controls.

The recently developed CRISPR-Cas9 genome editing technology[15–17] allows knockout (KO) of target genes and study of their biological functions in human cells. This straightforward and highly efficient approach is ideal for phosphoproteomic studies, as it greatly reduces the background. In the study described here, we combined the CRISPR-Cas9 technique and global quantitative phosphoproteomic analysis to discover new members in the AMPK-dependent signaling network. We generated AMPK-deficient HEK293A cells by doubly knocking out two functionally redundant AMPK catalytic subunits: AMPKα1 and AMPKα2. These function-deficient cells are ideal controls for global phosphoproteomic analysis. By using this procedure, we identified 109 phosphosites with markedly higher phosphorylation levels in HEK293A AMPK wild-type (WT) cells after AMPK activation than those in AMPKα1/α2-DKO cells. Another 51 phosphosites were found to be phosphorylated at lower levels in AMPK WT cells than those in AMPKα1/α2-DKO cells, suggesting that these are likely phosphorylation events that are negatively and probably indirectly regulated by AMPK expression. Further analysis of the 109 upregulated phosphosites using

known conserved AMPK phosphorylation motifs revealed 32 potential AMPK phosphorylation sites, 24 of which are newly discovered, previously unreported sites. We subsequently validated the phosphorylation site S45 of Armadillo repeat-containing protein 10 (ARMC10; alternative name SVH, specific splicing variant involved in hepatocarcinogenesis[18]) as an AMPK substrate site. Overexpression of ARMC10 promoted mitochondrial fission. Conversely, KO of ARMC10 prevented AMPK-mediated mitochondrial fission. Thus, we uncovered additional components of the AMPK-dependent signaling network and revealed ARMC10 as a novel AMPK substrate and effector involved in the dynamic regulation of mitochondrial fission and fusion.

## Results

**Comprehensive global phosphoproteomic screening for AMPK substrates.** To identify new AMPK substrates, we combined the CRISPR-Cas9 genome-editing technology and quantitative phosphoproteomic in our workflow (Fig. 1a). We designed guide RNAs (gRNAs) targeting AMPKα1 and AMPKα2, which are the two redundant catalytic subunits of the AMPK complex. We successfully generated AMPKα1/α2-DKO HEK293A-derived cells and confirmed the DKO by using Western blotting with anti-AMPKα1/α2 antibodies and genomic sequencing (Supplementary Figure 1a, b). We further functionally verified these AMPKα1/α2-DKO cells, finding that, in single AMPKα1-KO or AMPKα2-KO cells, phosphorylation at AMPKα T172 site could still be detected and increased further by treatment with the AMPK activator A769662. Downstream phosphorylation events, such as phosphorylation at the ACC1 S79 and ULK1 S555 sites, could also be detected and similarly increased in single subunit–KO cells. However, all of these phosphorylation events were abolished in AMPKα1/α2-DKO cells (Fig. 1b). Thus, these DKO cells with complete inactivation of AMPK activity were ideal background controls for our quantitative phosphoproteomic analysis to identify AMPK-dependent phosphorylation events.

Phosphorylation of Thr-172 in the activation loop of AMPKα is indicative of AMPK kinase activity. As shown in Fig. 1c, T172 phosphorylation during treatment with A769662 increased with time and was the greatest at the 24-h treatment time point. Similarly, while ACC1 phosphorylation was clearly detected at the 1-h time point, it also increased with time and reached its highest level at the 24 h time point. To maximize our ability to detect AMPK-dependent phosphorylation sites, we therefore treated both AMPK WT and AMPKα1/α2-DKO cells with A769662 100 μM for 24 h after stable isotope labeling by amino acids in cell culture (SILAC) (Fig. 1a). We combined these two cell samples with different SILAC labeling for use in the phosphoproteomic experiments. To enrich these phosphopeptides, we subjected the mixed-cell samples to trypsin digestion and captured phosphopeptides with $TiO_2$ beads and analyzed them by using an LTQ Orbitrap Elite mass spectrometer (ThermoFisher Scientific, Waltham, MA) (Fig. 1a). We conducted four biological repeats for phosphoproteomic comparison of WT and AMPKα1/α2-DKO cells, including two for "light-WT", "heavy-DKO", and two for reverse labeling (Supplementary Figure 1c).

We used the MaxQuant software program to search the raw mass spectrometry (MS) data and calculated the ratio between WT and AMPKα1/α2-DKO for the phosphopeptides[19]. In total, we acquired 9122 phosphopeptides (Supplementary Data 1). Of these phosphopeptides, 72.16% had a localization probability higher than our cutoff of 0.7 for phosphosite assignment. To obtain a high-confidence list of AMPK-regulated phosphopeptides, we used the data-filtration parameters of a fold-change greater than 1.5 and $p$ value < 0.01 ($t$-test) for the quantification

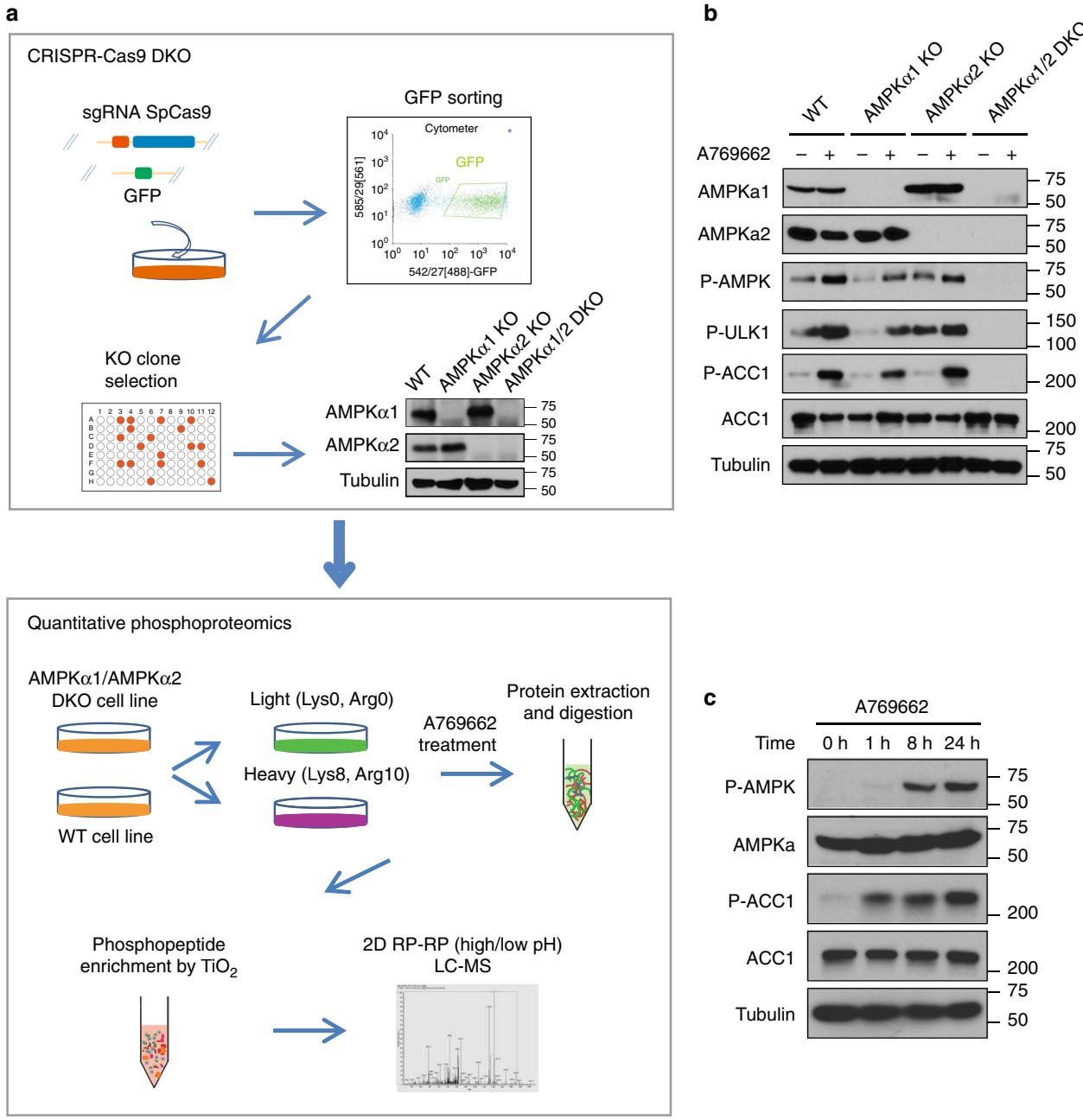

**Fig. 1** Phosphoproteomic screening to identify AMPK substrates. **a** The phosphoproteomics workflow. AMPKα1/α2–double knockout (DKO) HEK293A-derived cells were generated using the CRISPR-Cas9 genome-editing technology. These cells were used as controls to compare with parental (WT) cells with intact AMPK. SILAC-based quantitative phosphoproteomic analysis was performed to identify AMPK substrates. GFP green fluorescent protein, RP reverse phase, LC-MS liquid chromatograph-mass spectrometry, A769662 an AMPK activator. **b** Validation of AMPKα1/α2-DKO cells using anti-AMPK antibodies and antibodies recognizing known AMPK phospho-substrates. WT AMPK cells were used as a control. **c** Time course of the effect of treatment with the AMPK activator A769662 (100 μM) on WT HEK293A cells. Western blotting was conducted using antibodies as indicated. A treatment period of 24 h was selected because it led to maximal AMPK kinase activity as indicated by AMPKα phosphorylation at the Thr-172 site

of phosphopeptides. These analyses revealed 109 phosphosites with higher phosphorylation levels in WT cells than those in AMPKα1/α2-DKO cells (Fig. 2a, Supplementary Data 2). In addition, we uncovered 51 phosphosites with lower levels (fold-change <0.667 and *p* value < 0.01 by *t*-test) in WT cells than those in AMPKα1/α2-DKO cells (Fig. 2a, Supplementary Data 2).

To identify potential AMPK substrates in these global quantitative phosphoproteomic data, we used a conserved AMPK substrate motif to separate the 109 upregulated phosphosites into two groups (Fig. 2b). We first aligned our AMPK substrate

candidates with the AMPK consensus motif using multiple sequence alignment. We then manually examined the sequence alignment to determine whether each peptide was a candidate AMPK substrate (group 1), using the following rules: (1) if a peptide contained both major conserved sites, i.e., the P − 3 site as R/K/H and the P + 4 site as L/V/I, it was considered a candidate AMPK substrate; (2) if a peptide contained only one of the two major conserved sites, but also had any of the other three conserved sites—the P − 5 site as L/M/I, the P − 4 site as R/K, and the P + 3 site as N/D/E, it was also considered a candidate

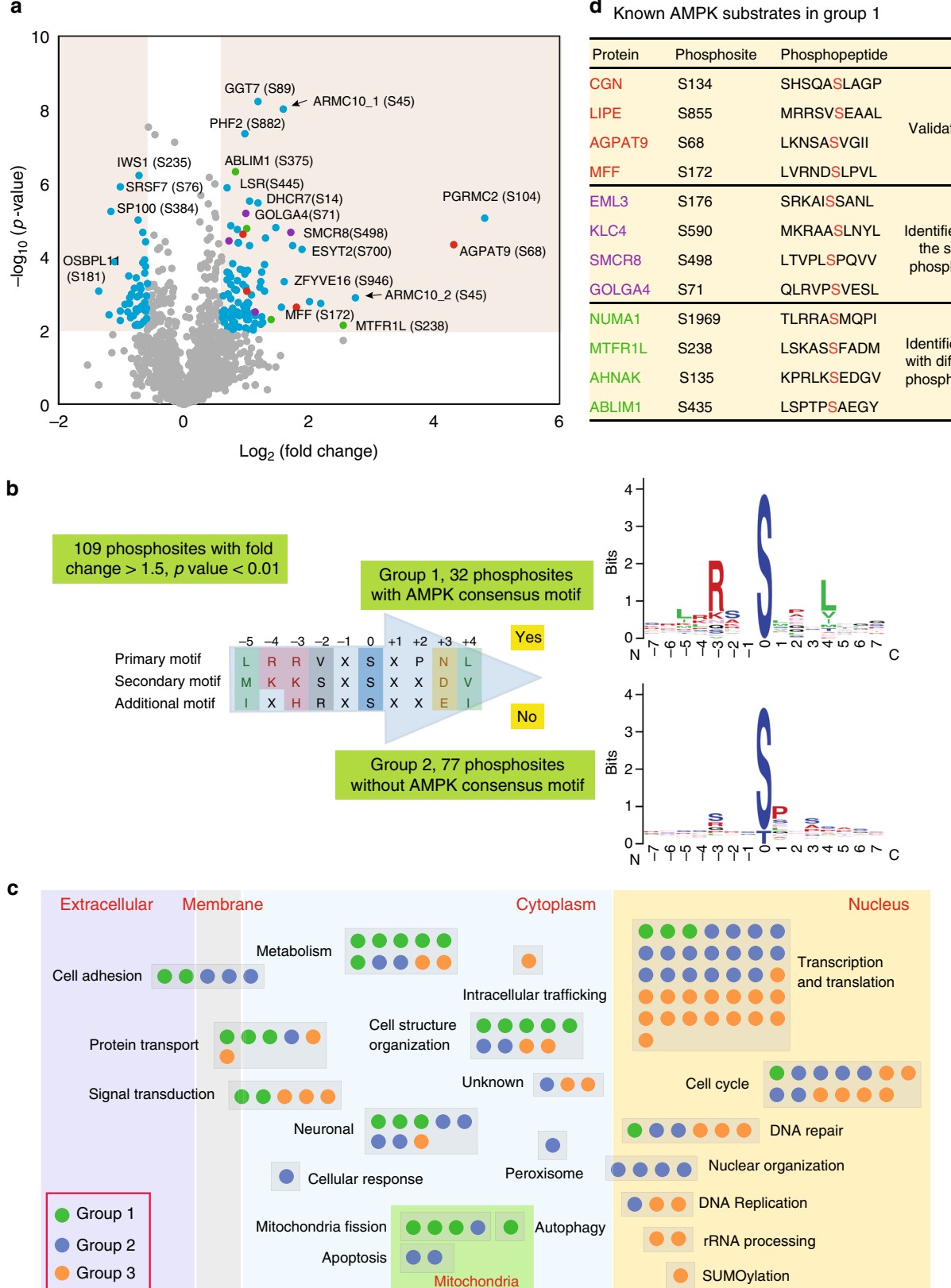

**d** Known AMPK substrates in group 1

| Protein | Phosphosite | Phosphopeptide | |
|---------|-------------|----------------|---|
| CGN | S134 | SHSQASLAGP | |
| LIPE | S855 | MRRSVSEAAL | Validated |
| AGPAT9 | S68 | LKNSASVGII | |
| MFF | S172 | LVRNDSLPVL | |
| EML3 | S176 | SRKAISSANL | |
| KLC4 | S590 | MKRAASLNYL | Identified with the same phosphosite |
| SMCR8 | S498 | LTVPLSPQVV | |
| GOLGA4 | S71 | QLRVPSVESL | |
| NUMA1 | S1969 | TLRRASMQPI | |
| MTFR1L | S238 | LSKASSFADM | Identified but with different phosphosite |
| AHNAK | S135 | KPRLKSEDGV | |
| ABLIM1 | S435 | LSPTPSAEGY | |

AMPK substrate. The phosphosites in group 1 ($n = 32$) (Supplementary Figure 2a) had peptide sequences similar to the conserved AMPK substrate motif[20] and may be sites directly phosphorylated by AMPK. The phosphosites that were upregulated in WT cells but did not meet these rules and did not have

sequences similar to the conserved AMPK substrate motif constituted group 2 ($n = 77$). These sites may be regulated by the AMPK-dependent signaling pathways but are not directly phosphorylated by AMPK. The 51 phosphorylation sites that had higher levels of phosphorylation in AMPKα1/α2-DKO cells than

**Fig. 2** Bioinformatics analysis of the phosphoproteomic results. **a** Volcano plot of quantitative analysis of phosphopeptides identified by MS. Peptides with fold-changes (WT AMPK:AMPKα1/α2-DKO ratio) >1.5 (p < 0.01, t-test) were selected as peptides phosphorylated at markedly higher levels in WT cells than in AMPKα1/α2-DKO cells. Red dots are the AMPK phosphorylation substrates that had previously been validated; purple dots are peptides identified as AMPK phosphorylation substrates but not validated; green dots are peptides identified as AMPK phosphorylation substrates by previous study, but not the same phosphorylation sites; blue dots are other AMPK phosphorylation substrates identified by our experiments; gray dots are the phosphorylation peptides identified by this study but showing no significant difference between AMPKα1/α2-DKO and WT samples. The black arrows indicate the phosphopeptides from ARMC10, which we studied further. Please note that these two phosphopeptides are from the same S45 phosphorylation site in two different ARMC10 isoforms (i.e., ARMC10_1 is in a shorter isoform of 308 residues and ARMC10_2 is in a longer isoform of 343 residues). **b** Logo motif of the phosphosites. The 109 phosphosites phosphorylated at higher levels in WT AMPK than in AMPKα1/α2-DKO cells were separated into groups 1 (with the conserved AMPK substrate motif) and 2 (without the motif). The phosphosites in these two groups were further analyzed by using WebLogo to create the sequence logos. **c** Bioinformatic analysis of 160 AMPK-regulated phosphosites. The 109 phosphorylated proteins in groups 1 or 2, as well as the 51 in group 3 (phosphorylated at significantly lower levels in WT AMPK cells than in AMPKα1/α2-DKO cells) were analyzed individually for their potential functions with Ingenuity Pathway Analysis and reference mining. Each dot in the figure represents one protein. **d** Known AMPK substrates in group 1. These substrates comprised three categories: validated, identified with high-throughput methods but not validated and identified as a substrate but with a different phosphosite

in WT cells were classified as group 3. For these phosphosites, the motif analysis identified only one conserved amino acid P at the site right after the phosphosite, which is a potential phosphorylation site for cyclin/cdks and/or MAPK (Supplementary Figure 2b). These data probably agree with the functions of AMPK, which prevent cell proliferation and/or cell cycle progress to reduce energy consumption.

We conducted functional analysis of these three groups of proteins using Ingenuity Pathway Analysis and reference mining (Fig. 2c, Supplementary Figure 2c–e, and Supplementary Data 3). Group 1 are direct AMPK substrate candidates. These genes/proteins mainly function in cytoplasm, including multiple functions such as metabolism, cell structure organization, and mitochondrial fission. Group 2 and group 3 are genes/proteins that may be indirectly regulated by AMPK. The majority of their functions are in the nucleus, including transcription and translation, cell cycle and DNA repair as the top three functions.

**Validation of potential AMPK substrates**. Of the 32 phosphosites identified as candidate AMPK substrates by our quantitative phosphoproteomic analysis, four were reported in previous studies as AMPK phosphorylation sites (Fig. 2d). Specifically, they are sites in MFF S172[11], CGN S137[14], LIPE S855[21], and AGPAT9 S68[13]. Another four of these phosphosites were previously identified by high-throughput strategies but have not yet been validated (Fig. 2d)[12–14]. An additional four of the proteins had been shown to be regulated by AMPK, but were revealed by our analysis to be new phosphosites (Fig. 2d)[12–14], including NUAK1 S1969 (reported at S1853), MTFR1L S238 (reported at S103), AHNAK S135 (reported at S572), and ABLIM1 S706 (reported at S450|S452). For these four AMPK-dependent substrates, we identified different phosphosites than that previously reported. This may indicate that multiple AMPK phosphorylation sites exist in these proteins. These four sites and the remaining 20 sites are potentially novel AMPK phosphorylation sites, as indicated by our phosphoproteomic and bioinformatics analysis (Supplementary Data 4).

In our data set, we identified a KLC family protein (KLC4) that is potentially phosphorylated by AMPK at S590, which agrees with the results of a previous high-throughput study[12]. An early study suggested that AMPK-dependent phosphorylation of another protein in this family (KLC2) at S539 and S582 is required for the association of the KLC interaction scaffolding protein 14-3-3[22], and it is crucial for PI3K transportation[23]. Because the protein sequence identity between KLC2 and KLC4 is 67.19%, and the sequence surrounding the KLC4 S590 site (i.e., MKRAApSLNYLN) is very similar to the sequence surrounding the KLC2 S582 site (i.e., MKRASpSLNFLN), we speculate that AMPK-dependent phosphorylation of KLC4 may participate in a similar or related function.

To experimentally validate additional novel AMPK substrates, we chose five sites (ARMC10 S45, REEP1 S150, REEP2 S152, KLC4 S590, and MFF S172) in group 1 for further analysis. ARMC10 S45 is at the top of the list when ranked by p value. Furthermore, we identified two different peptides for ARMC10 S45, which differ only by an alternative splicing site two residues after the phosphorylation site. REEP1 S152 and REEP2 S150 are from the same protein family, and the phosphorylation sites are within the same region with high sequence similarity (REEP1 S152 site: RLRSFpSMQDL; REEP2 S150 site: KLRSFpSMQDL). As already mentioned, the KLC4 S590 site is conserved and shares extensive sequence similarity with known AMPK phosphorylation site S582 in KLC2. MFF S172 is a control, which was reported as an AMPK phosphorylation site[11]. The six peptides for these five phosphorylation sites (ARMC10 S45_1 and ARMC10 S45_2 are two peptides but the same phosphorylation site from different isoforms) consistently showed stronger signals in AMPK WT cells than in AMPKα1/α2-DKO cells (Fig. 3a).

The cDNAs of these genes were fused with S protein, Flag, streptavidin-binding peptide (SFB) tags and transiently expressed in AMPKα1/α2-DKO or WT cells. We harvested cells after treatment with 100 μM A769662 for 24 h, pulled down the overexpressed proteins with streptavidin beads, and analyzed them via Western blotting with an anti-phospho-Ser antibody and phospho-AMPK substrate motif antibody, respectively. We found that signals for ARMC10, KLC4, and REEP2 were higher in AMPK WT cells than in AMPKα1/α2-DKO cells as detected using the anti-phospho-Ser antibody (Fig. 3b). We did not detect REEP1, probably because of the relatively low sensitivity of this phospho-Ser antibody for Western blotting. We also failed to detect MFF phosphorylation using this antibody. Besides the sensitivity of the phospho-Ser antibody, the expression level of MFF was low when compared to other tested genes/proteins. We found that the phospho-AMPK substrate motif antibody, which is more specific than the phospho-Ser antibody, was able to detect MFF phosphorylation. However, the phospho-AMPK substrate motif antibody detected only the phosphorylated MFF (Fig. 3b). This may reflect the specificity of this antibody, which recognizes the sequence LXRXX(S/T), which appears only in the MFF protein. We further used this AMPK substrate motif antibody with several other known AMPK substrates, such as ACC1 S79 and ULK1 S555, and could not detect any signals by Western blotting (Supplementary Figure 3), indicating that this antibody can be used to detect only a subset of AMPK substrates.

**ARMC10 is a novel AMPK substrate**. Our phosphoproteomics study identified the ARMC10 phosphorylation site S45 with two different peptides, both of which had considerably higher

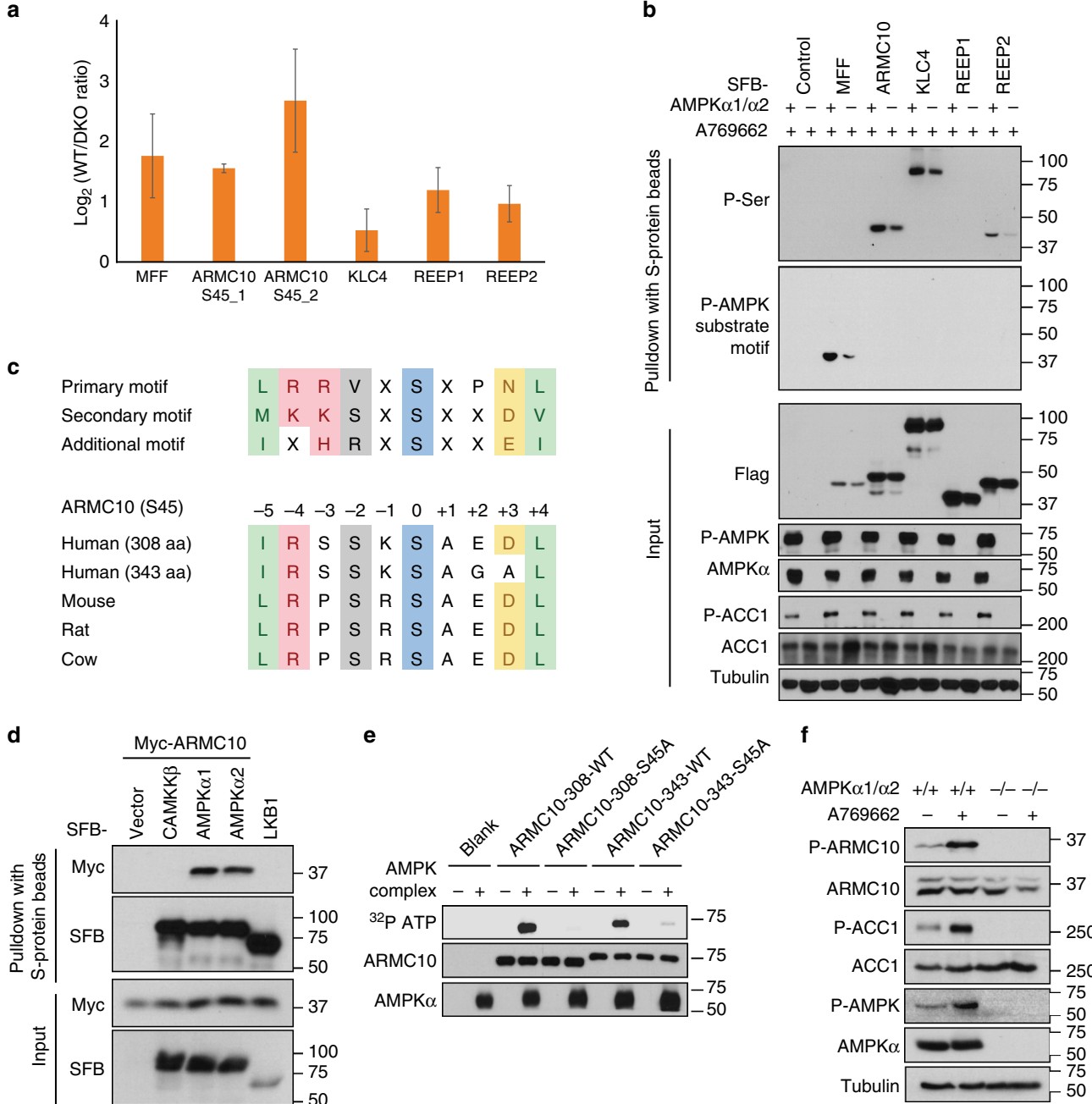

**Fig. 3** Validation of AMPK substrates. **a** Phosphoproteomic quantification of the five candidate AMPK phosphorylation sites selected for validation. ARMC10 (S45)_1 is the phosphopeptide from the shorter isoform of ARMC10 (308 residues). ARMC10 (S45)_2 is the phosphopeptide from the longer isoform of ARMC10 (343 residues). **b** Validation of the five selected AMPK substrates. For this experiment, constructs encoding SFB-tagged genes were transfected into WT or AMPKα1/α2 HEK293A cells. Cells were treated 16 h later with AMPK activator A769662 (100 μM) for 24 h. Cells were collected and subjected to lysis in NETN cell lysis buffer. Cell lysates were subjected to pulldown assay using S protein beads. Western blotting was conducted using indicated antibodies. **c** Conservation of the phosphorylation site in ARMC10 in mammals. **d** Association of ARMC10 with AMPK. Constructs encoding Myc-tagged ARMC10 and indicated constructs for SFB empty vector, SFB-tagged AMPKα1, SFB-tagged AMPKα2, or SFB-tagged control genes were co-transfected into HEK293T cells. The two control genes, CAMKKβ and LKB1, are two kinases that phosphorylate AMPKα and regulate its function, but they normally do not share substrates with AMPK. Cell lysates were subjected to pulldown assays with S protein beads and analyzed by Western blotting using the indicated antibodies. **e** Both isoforms of ARMC10 were phosphorylated by the AMPK complex at the S45 site. In vitro kinase assays were performed with γ-$^{32}$P-ATP, and PhosphorImager software was used to detect the kinases. **f** In vivo analysis of ARMC10 phosphorylation. Anti-phospho-S45 ARMC10 antibody was used to detect changes in the phosphorylation of ARMC10 S45 before and after treatment with the AMPK activator A796662 100 μM for 24 h in HEK293A WT or AMPKα1/α2-DKO cells

phosphorylation levels in AMPK WT cells than those in AMPKα1/α2-DKO cells (Fig. 3a). These two peptides are derived from the same phosphorylation site in two isoforms of ARMC10 with alternative splicing at the +2 position of the S45 site. Both sites fit well with the conserved AMPK substrate motif and are highly conserved in vertebrates (Fig. 3c). Given the strong evidence for this phosphosite in our proteomic study and our validation results shown in Fig. 3b.

To determine whether phosphorylation of the ARMC10 S45 site is regulated by AMPK, we first confirmed the specific interaction between AMPK and ARMC10 by pulldown assay. ARMC10 interacted only with AMPKα1 and AMPKα2, not with LKB1 or CAMKKβ (Fig. 3d). The two phosphopeptides we identified for the ARMC10 S45 site are the same phosphorylation site in two different ARMC10 isoforms: a shorter isoform of 308 residues and a longer isoform of 343 residues. We generated the WT and ARMC10 S45A mutants for both isoforms, and in vitro kinase assays demonstrated that AMPK could directly phosphorylate ARMC10 at S45 (Fig. 3e). Next, we generated a S45 phospho-specific antibody and confirmed the in vitro kinase assay results via immunoblot analysis using this antibody (Supplementary Figure 4a). An in vitro kinase assay of overexpressed WT ARMC10 or ARMC10 S45A mutant purified from human HEK293T cells also verified the phosphorylation and regulation of ARMC10 by AMPK (Supplementary Figure 4b). With the ARMC10 S45 phospho-antibody, we can only detect ARMC10 phosphorylation in WT ARMC10 transfected cells, but not in S45A mutant transfected cells (Supplementary Figure 4c). Moreover, phosphorylation of ARMC10 at S45 increased in cells treated with A769662 (Supplementary Figure 4d), and this phosphorylation could be detected only in AMPK WT cells, not in AMPKα1/α2-DKO cells (Fig. 3f), suggesting that this site is indeed phosphorylated in vivo by AMPK.

We detected phosphorylation only in the short isoform of ARMC10 (Fig. 3f and Supplementary Figure 4d). This may have been due in part to the longer isoform being much less abundant than the shorter isoform (Fig. 3d and Supplementary Figure 4d). In addition, the phospho-antibody was generated using the peptide GIRSSKpSAED derived from the dominant short isoform. The longer isoform has a slightly different sequence (i.e., GIRSSKpSAGA). Therefore, the phospho-specific antibody may prefer to recognize the phosphosite of the short isoform. Because the short form of ARMC10 is the dominant isoform in human cells (Fig. 3f and Supplementary Figure 4d, e) and is the only form existing in other species (Fig. 3c), we used it in our follow-up studies.

**ARMC10 overexpression induces mitochondrial fission.** ARMC10 is not a well-studied protein, and information about this gene product is very limited. Engineered ascorbate peroxidase molecular labeling demonstrated that ARMC10 was located in the outer membrane of mitochondria, with most of the protein in the cytosol[24]. We confirmed the mitochondrial localization of ARMC10 in HEK293T by using immunostaining (Fig. 4a) and found that its colocalization with the mitochondrial marker TOM20 was dependent on its transmembrane domain (Supplementary Figure 5a), which is located at residues 5–27 at the N-terminus of ARMC10. Moreover, overexpression of full-length ARMC10 seemed to change the shape of the mitochondria. To determine whether ARMC10 expression−induced changes in mitochondrial morphology are affected by the AMPK-dependent phosphorylation of ARMC10, we induced transient expression of SFB-tagged WT ARMC10 or ARMC10 S45A mutant in HEK293T cells. Cells overexpressing WT ARMC10 showed drastic changes in mitochondrial morphology compared with

those overexpressing the ARMC10 S45A mutant (Fig. 4a). Quantitative analysis revealed that the percentages of short and fragmented mitochondria were significantly higher in cells over-expressing WT ARMC10 than in cells overexpressing ARMC10 S45A mutant (Fig. 4b), suggesting that ARMC10 may be involved in AMPK-dependent regulation of mitochondrial dynamics.

**AMPK substrate ARMC10 regulates mitochondrial dynamics.** Changes in mitochondrial morphology indicate a potential influence on mitochondrial fusion and fission[25]. AMPK is a master regulator of mitochondrial homeostasis, which couples mitochondrial fission with mitophagy after prolonged energy stress and signals to the nucleus to initiate biogenesis of new mitochondria to replace the damaged ones[26–28]. This regulation of mitochondrial homeostasis requires AMPK activity and is inhibited in AMPKα1/α2-DKO cells that have lost both AMPKα1 and AMPKα2[29,30]. A previous study suggested that MFF is a direct substrate of AMPK that participates in regulation of mitochondrial fission[11]. Here, we propose that ARMC10 may also function in the regulation of mitochondrial fission.

We showed that ARMC10 can be directly phosphorylated by AMPK at S45 and that ARMC10 overexpression promotes mitochondrial fission. Given that AMPK activity is required for facilitating mitochondrial fission, we examined whether it could be regulated by AMPK-dependent phosphorylation of ARMC10 at S45. We induced transient expression of WT ARMC10 or the ARMC10 S45E mutant, which mimics the S45 phosphorylated form of ARMC10, in WT AMPK and AMPKα1/α2-DKO HEK293A cells and analyzed their mitochondrial morphology via immunostaining (Fig. 4c, Supplementary Figure 5c). Quantification revealed that the degree of mitochondrial fission was significantly higher in WT AMPK cells treated with A769662 than in AMPKα1/α2-DKO cells (Fig. 4d, Supplementary Figure 5d). However, unlike cells expressing WT ARMC10, both WT AMPK and AMPKα1/α2-DKO cells expressing the ARMC10 S45E mutant, with or without AMPK activation, had greater mitochondrial fission. However, the levels of mitochondrial fission in cells expressing ARMC10 S45E mutant were still less than that in WT cells, which may due to the possibility that the ARMC10 S45E mutant is less penetrant in the absence of AMPK directed phosphorylation of MFF. Another non-exclusive explanation is that the ARMC10 S45E mutant may only partially restore the function of ARMC10. Nevertheless, these findings suggest that AMPK-dependent phosphorylation of ARMC10 at S45 is an important effector of AMPK-mediated mitochondrial fission.

To further determine the roles of ARMC10 in the control of mitochondrial dynamics, we generated ARMC10-KO U2OS cells by using the CRISPR-Cas9 technology (Supplementary Figures 4e, f). WT ARMC10 and ARMC10-KO U2OS cells were treated or not treated with A769662. Immunofluorescent staining indicated that mitochondrial fission increased drastically in WT ARMC10 U2OS cells after AMPK activation, but this did not occur in ARMC10-KO cells. The increased mitochondrial fission could be rescued by overexpression of ARMC10 WT or S45E mutation, but not ARMC10 S45A mutation (Fig. 5a, b). These results again indicate that ARMC10 has an important role in mitochondrial fission, a process regulated by AMPK.

**Phosphorylation of ARMC10 by AMPK is important for cell survival.** Mitochondrial fission and fusion have been implicated in cell death. Previous studies suggest that mitochondrial fission leads to apoptosis, whereas mitochondrial fusion is antiapoptotic[31,32]. We compared the proliferation of WT, AMPKα1/α2-DKO, and ARMC10-KO in HEK293A- or U2OS-

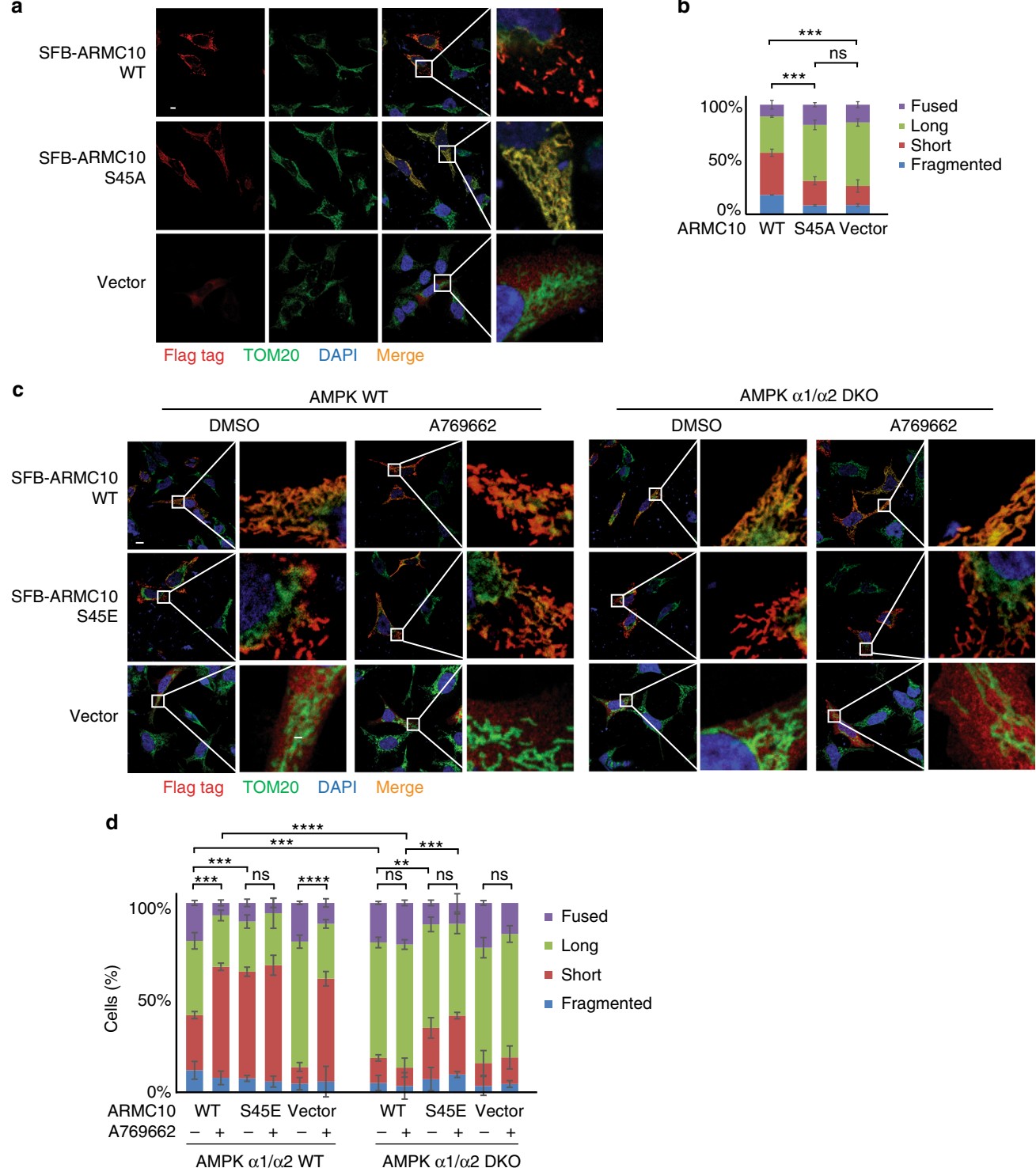

derived cells after glucose starvation; only the WT cells survived robustly, whereas ARMC10-KO cells behaved the same as AMPKα1/α2-DKO cells, with significant reduction in cell survival (Fig. 5c, d). Notably, the cell survival defects in ARMC10-KO cells could be rescued by the stable expression of WT ARMC10 but not by the ARMC10 S45A mutant (Fig. 5c, d).

ARMC10 can induce changes in mitochondrial morphology and alter mitochondrial distribution, which may also cause mild mitochondrial aggregation[33]. We found that a small fraction of U2OS cells had this mitochondrial aggregation phenotype after transient expression of ARMC10 and treatment with A769662

(Supplementary Figure 5b). Previous studies described these progressive changes in mitochondrial morphology[34,35], in which mitochondria first exhibited fission and then aggregation, followed in some cases by mitophagy. To determine whether ARMC10 is involved in these mitochondrial transitions, we examined microtubule-associated protein 1 light chain 3 (LC3) staining as a marker of autophagy. As shown in Supplementary Figures 6a, b, punctate LC3 staining increased after treatment with the AMPK activator A769662. However, only in ARMC10 WT cells, the percentage of LC3 puncta colocalized with the mitochondrial marker TOM20 increased significantly. Notably, in

**Fig. 4** Function assay of ARMC10 and ARMC10 phosphorylation at S45. **a, b** Analysis of mitochondrial morphology in HEK293A cells overexpressing ARMC10. Transient expression of wild-type (WT) ARMC10 and the ARMC10 S45A mutant was confirmed via immunostaining. The transfection with vector only was included as the control. In (**a**) a region of the cell with ARMC10 WT or S45A mutation overexpression was enlarged to show the mitochondrial shape. The green signal indicates the mitochondrial marker TOM20, the red signal indicates anti-Flag staining for WT ARMC10 or the ARMC10 S45A mutant, and the blue signal indicates 4′,6-diamidino-2-phenylindole (DAPI)/nuclei. Scale bar: 10 μm. In (**b**) quantification of the mitochondrial morphology following expression of WT ARMC10 or the ARMC10 S45 mutant. Data are means ± standard error of the mean from three independent experiments with 150 cells for each replicate. The quantification was done by a person who was blinded to genotype and treatment of the samples. The mitochondria were characterized as follows: "fragmented", the majority of mitochondria were spherical; "short", the majority of mitochondria were less than ~10 μm; "long", the majority of mitochondria were more than ~10 μm; and "fused" was highly interconnected mitochondria with <4–5 free ends. We used long versus short morphology for comparison in our statistical analysis. ***$p < 0.001$ and ****$p < 0.0001$ (two-way analysis of variance using the Tukey multiple comparison test)[11]; ns not significant. **c, d** Analysis of mitochondrial morphology in parental WT AMPK and AMPKα1/α2-DKO HEK293A cells. WT ARMC10 or the phospho-mimicking ARMC10 S45E mutant was transiently expressed in WT and AMPKα1/α2-DKO cells. In (**c**) immunostaining experiments were conducted using these cells treated with AMPK activator A769662 300 μM or dimethyl sulfoxide (DMSO) for 1 hour. The green signal indicates the mitochondrial marker TOM20, the red signal indicates anti-Flag staining for WT ARMC10 or the ARMC10 S45E mutant, and the blue signal indicates DAPI/nuclei. Scale bar: 10 μm. A region of the cell with ARMC10 WT or S45E overexpression was enlarge to show the mitochondrial shape. **d** Quantification of the mitochondrial morphology in cells shown in (**c**). Morphology was quantified as for Fig. 4b

ARMC10-KO cells, we did not observe colocalization of LC3 with TOM20 changed after treatment with A769662, indicating that ARMC10 may have an important role not only in AMPK-induced mitochondrial fission but also in its transition to mitophagy.

**Interactome study suggests a potential mechanism of ARMC10 in mitochondrial dynamics**. To further explore the potential mechanism of ARMC10 function in mitochondrial regulation, we performed an interactome study of this protein by using a BioID technique[36]. We generated stable BioID2-tagged ARMC10-expressing HEK293T cells lines and performed BioID2-MS experiments with these cell lines six times. We used two controls—a BioID2 tag only and a BioID2 tag fused with another protein—for comparison. The candidate ARMC10 interactome comprised many proteins involved in mitochondrial fission and mitophagy (Fig. 6a). These proteins include DNM1L (also called Drp1), a protein critically involved in mitochondrial fission[37]; MFF, MIEF1, and MIEF2, which regulate the recruitment of Drp1 to mitochondria[38]; MTFR1 and MTFR2, which are mitochondrial fission regulators[39]; and BNIP3 and BAX, which mediate mitochondrial dysfunction and mitophagy[40].

We selected four well-known mitochondrial fission or fusion genes (MFF, FIS, MFN1, and OPA1) for pulldown experiments to test for potential interactions with ARMC10 in the regulation of mitochondrial dynamics. We found that ARMC10 can interact with two mitochondrial fission proteins, MFF and FIS1, that are important in mediating Drp1 recruitment during mitochondrial fission[41], but no obvious binding was noted between ARMC10 and the two mitochondrial fusion proteins MFN1 and OPA1 (Fig. 6b). We further confirmed the binding between ARMC10 and MFF (Supplementary Figures 6c, d). As MFF and FIS1 can bind to and recruit Drp1 during mitochondrial fission, we wondered whether ARMC10 has a similar function. Indeed, as shown in Fig. 6c, we found that ARMC10, like MFF, binds to Drp1. These findings suggest that ARMC10 may participate in mitochondrial fission by interacting with several proteins involved in this process, including MFF, FIS1, and Drp1 (Fig. 6d).

## Discussion

In the present study, we performed global quantitative phosphoproteomic analysis and discovered 160 AMPK-dependent phosphorylation sites, and 32 of these sites are likely direct AMPK phosphorylation sites. We validated one uncharacterized protein, ARMC10, and demonstrated that the S45 site of ARMC10 can be phosphorylated by AMPK. According to our study, ARMC10 is an effector of AMPK that participates in dynamic regulation of mitochondrial fission and fusion in response to energy stress.

Identifying kinase substrates is critical to understanding the functions of a given protein kinase. However, this is an extremely challenging task because of the lack of a universal approach for discovery of kinase substrates. Previous studies have used several well-designed and specific strategies to identify AMPK substrates, including pulldown assays with a specific anti-AMPK motif antibody or 14-3-3 protein and comparisons of samples treated or not treated with an AMPK activator or inhibitor. These strategies worked well and led to the discovery of more AMPK substrates with diverse cellular functions[9,10,12,14]. With these strategies, however, the basal activity of endogenous AMPK could increase the background signals, thereby complicating the identification of bona fide AMPK-dependent phosphorylation events. The well-developed CRISPR-Cas9 genome-editing technology greatly facilitates our ability to disrupt a target gene and thus completely abolish the function of this target protein. Therefore, we combined the CRISPR-Cas9 technology and global quantitative phosphoproteomic in this study to set up a workflow for the identification of AMPK substrates. We successfully discovered 32 phosphosites regulated by AMPK kinase, 24 of which are novel, indicating that the combination of CRISPR-Cas9−mediated KO cells and global quantitative phosphoproteomic is suitable as a general approach for the identification of kinase substrates. Moreover, this strategy does not require specific phosphorylation motif antibodies or any extensive knowledge of the particular kinase under investigation. We will continue to use this general approach to study kinase-dependent signaling pathways, especially the ones that have not yet been studied extensively.

There are some drawbacks to this approach for the discovery of kinase phospho-substrate relationship. First, it is a global phosphoproteomics approach with only $TiO_2$ enrichment, which enriches all the phosphorylated peptides. The phosphosites regulated by the particular kinase of interest comprise just a very small fraction of the global phosphoproteome. Many of the phospho-substrates will be missed in such screens because of the relatively limited coverage of current phosphoproteomics. For example, we identified eight known AMPK substrates but missed many other well-known AMPK substrates. Second, the strategy of comparing kinase KO cells with WT cells may have intrinsic problems. KO of a particular protein kinase may cause compensation and/or cellular stress, which in theory can lead to some unpredictable events in the cell that may impede the discovery of true physiological substrates of the kinase of interest. Third, since the strategy we use is a global phosphoproteomics approach,

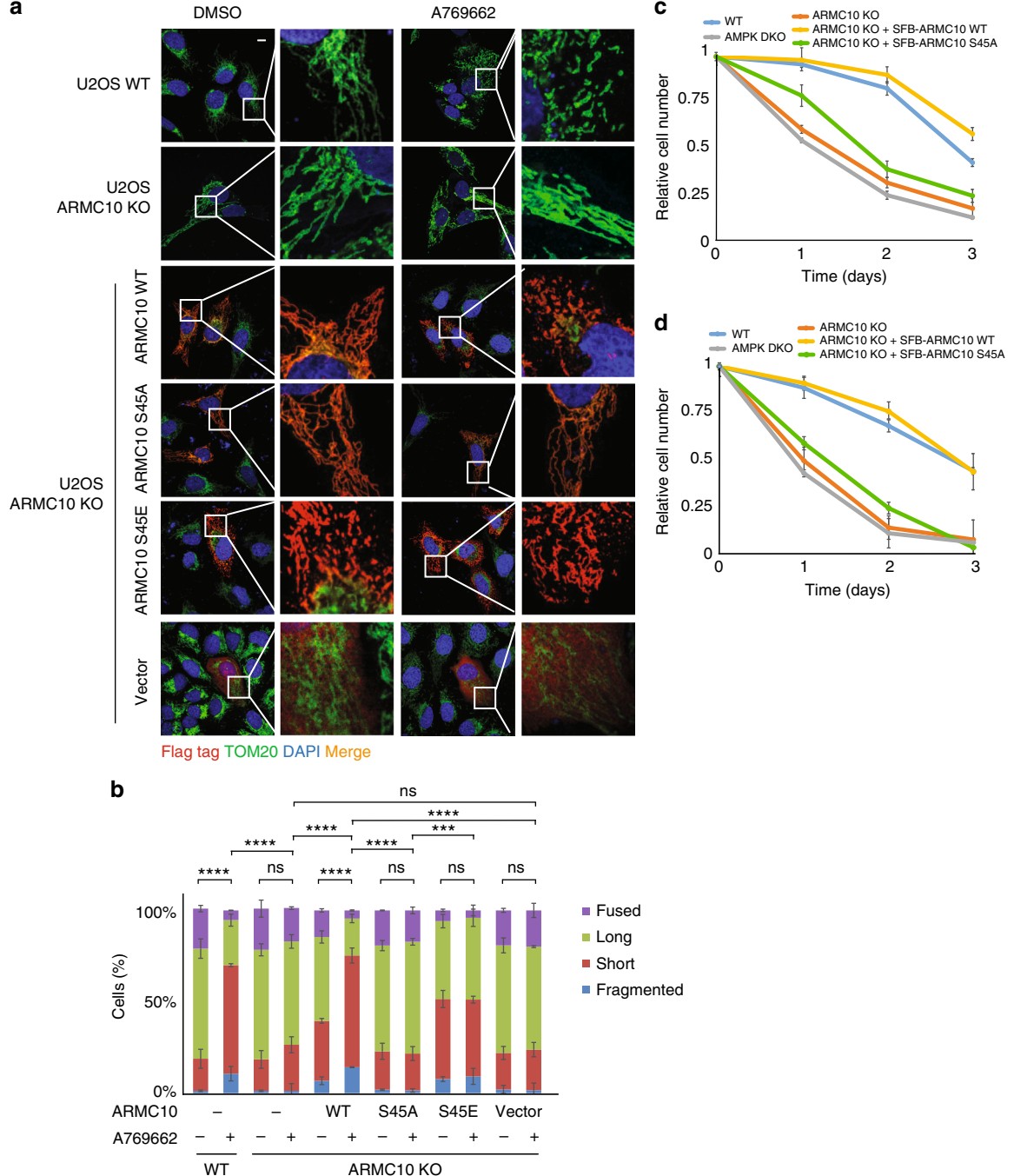

**Fig. 5** Mitochondrial morphology changes in ARMC10-KO cells. **a** Results of immunostaining experiments using wild-type (WT) ARMC10 or ARMC10-KO U2OS cells treated or not treated with AMPK A769662 300 μM for 1 hour. Dimethyl sulfoxide (DMSO) was used as control. The rescue experiments were done with transiently expressed ARMC10 WT, S45A mutation, and S45E mutation. The green signal indicates the mitochondrial marker TOM20, and the blue signal indicates DAPI/nuclei. Scale bar: 10 μm. **b** Quantification of mitochondrial morphology in WT ARMC10 and ARMC10-KO U2OS cells. Morphology was quantified as for Fig. 4b, d. **c**, **d** Cell survival following glucose starvation. HEK293A-derived cells (**c**) or U2OS-derived cells (**d**) were cultured in 25 mM glucose-containing medium for 24 h and then changed to a medium without glucose. Cells were counted at each time point (mean ± standard deviation; $n = 3$ biologically independent extracts) and plotted

many of the phosphopeptides with higher phosphorylation levels in WT but not in KO cells are not direct substrates of the kinase but are indirectly regulated by the kinase. We used the well-known conserved AMPK substrate motif to identify the ones that may be directly phosphorylated by AMPK. But for many kinases without any known substrate motif, it will be challenging to pick out the real direct kinase substrates. Fourth, our approach relies on the generation of kinase deficient/KO cells, which will not

work for many protein kinases that are essential for cell survival. For these essential kinases, the alternative approaches are the use of knockdown cells or kinase inhibitors as the control, but these alternatives are not as efficient as KO since they cannot completely block the kinase activity.

Our functional analysis of the three groups of proteins regulated by AMPK confirmed the diverse functions of this kinase, especially its roles in metabolism and signaling pathways.

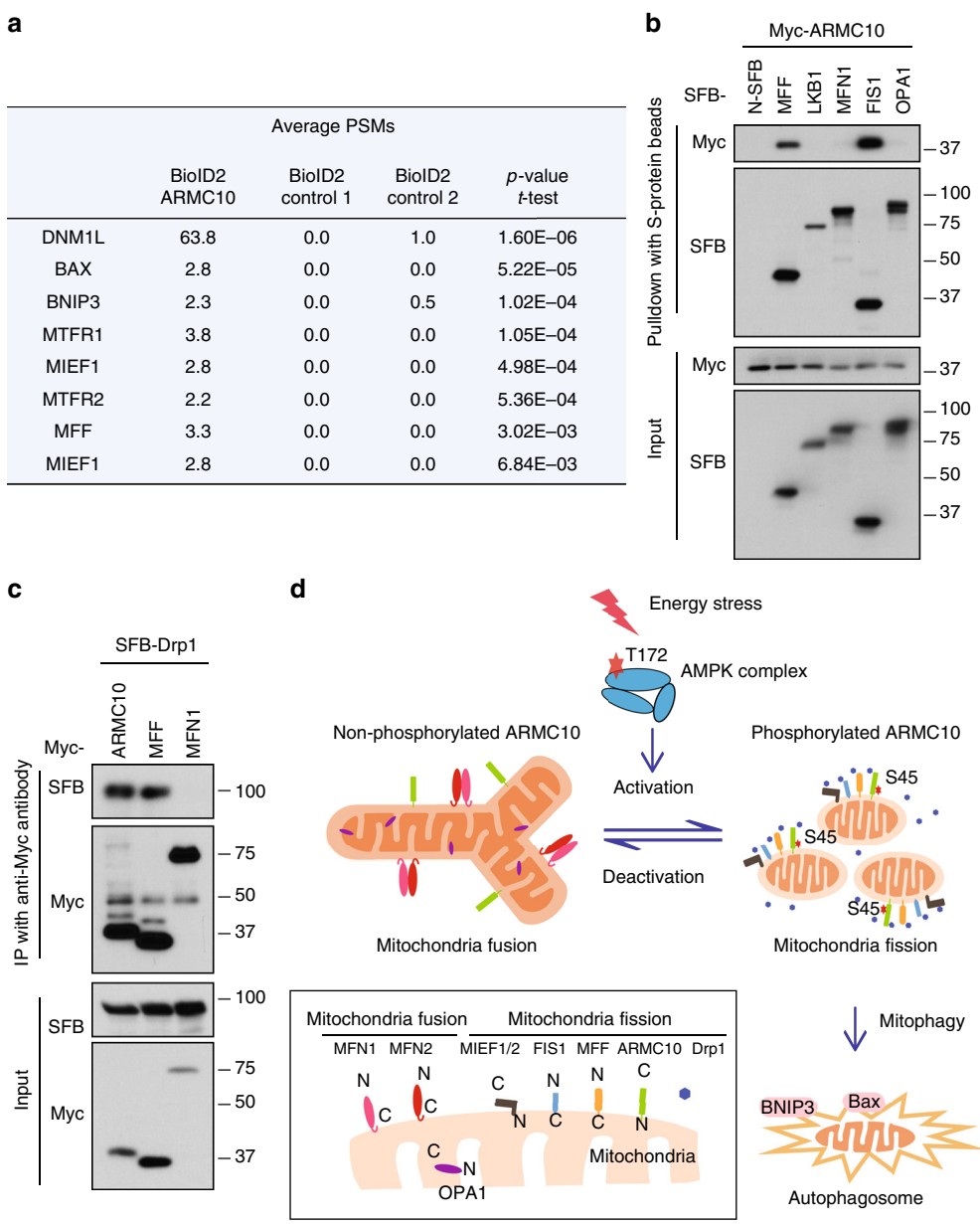

**Fig. 6** ARMC10 associates with mitochondrial fission proteins. **a** Mitochondrial fission proteins enriched by BioID pulldown with the bait protein ARMC10. Six repeats with the BioID2-tagged ARMC10 and eight controls. Control 1, BioID2 tag fused with mitochondria localization sequence (two repeats); control 2, BioID2 tag fused with a nonrelated bait protein, SLX4IP (six repeats), were conducted for the BioID experiments. The peptide spectrum matches (PSM) presented are the average numbers of PSMs of the candidate interacting genes involved in mitochondrial fission and/or mitophagy identified in ARMC10 or control groups. The p values represent statistical analysis of identified PSMs of the BioID2-tagged ARMC10 binding proteins comparing to those identified in all control samples. **b** ARMC10 associates with mitochondria fission proteins but not mitochondria fusion proteins. Mitochondria fusion (MFN1 and OPA1) and fission (MFF and FIS1) proteins were selected and tested for binding to ARMC10. N-terminal SFB vector alone (N-SFB) and SFB-tagged LKB1 were selected as negative controls. For these experiments, constructs encoding indicated tagged proteins were co-transfected into HEK293T cells. Cells were collected 24 h later and cell lysates were subjected to pulldown assay using S protein beads. Western blotting was conducted using indicated antibodies. **c** ARMC10 binds to Drp1. Constructs encoding Myc-tagged ARMC10, MFF, and MFN1 were co-transfected with constructs enconding SFB-tagged Drp1. The association of ARMC10 or MFF with Drp1 was tested as described for (**b**) with immunoprecipitation (IP) by anti-Myc antibody and Western blotting using indicated antibodies. **d** A working model of AMPK-regulated ARMC10 S45 phosphorylation in mitochondrial fission. In response to energy stress, AMPK is activated by its phosphorylation at T172. The activated AMPK phosphorylates ARMC10 at the S45 site. The phosphorylated/activated ARMC10 works with several mitochondrial fission proteins, such as MFF, Drp1, FIS1, and MIEF1/2, to promote mitochondrial fission. If fission continues, cells may undergo mitophagy. When AMPK is deactivated, mitochondria will undergo fusion with the help of several mitochondria fusion proteins, including MFN1, MFN2, and OPA1

Notably, our study revealed several mitochondrial proteins, such as MFF, MTFR1L, and ARMC10, to be AMPK substrates. Indeed, Shaw and colleagues recently reported that MFF is an AMPK substrate that functions in the regulation of mitochondrial fusion and fission[11,42]. Two different AMPK substrate screening studies identified MTFR1L[12,13], indicating that MTFR1L is also likely a direct AMPK substrate. In this study, we validated ARMC10 as a newly discovered AMPK substrate that participates in

mitochondrial fission. Thus, AMPK may have important roles in regulating mitochondrial dynamics via phosphorylating several components in this process. Its functions in mitochondria regulation need to be further elucidated.

We identified a number of proteins with phosphosites that differed considerably between control and AMPKα1/α2-DKO cells but did not have the conserved AMPK substrate motif in their protein sequences (groups 2 and 3). We speculated that these proteins are indirectly regulated by AMPK. By using bioinformatics functional analysis, we found that these phosphoproteins have highly enriched nuclear functions. Many of them function in transcriptional and translational regulation. Groups 2 and 3 are also enriched in cell-cycle progression, DNA repair and nuclear organization. We suspect that phosphorylation of these nuclear proteins, which may be indirectly regulated by AMPK, enables AMPK to participate in the regulation of transcriptional and translational processes and thus ensure energy balance by increasing the capacity of cells to produce ATP but reduce ATP consumption. AMPK signaling may also promote cell survival via inhibition of cell-cycle progression and cell proliferation.

From our list of candidate AMPK substrates, we identified two ARMC10 phosphopeptides for the same phosphorylation site, S45, strongly suggesting that ARMC10 is phosphorylated in vivo by AMPK at this site. The results of the subsequent functional studies suggest that ARMC10 is likely an important AMPK substrate involved in the regulation of mitochondrial functions. Moreover, we showed that ARMC10 is a key mediator involved in AMPK-dependent regulation of mitochondrial fusion and fission, and it relies on AMPK and AMPK-dependent phosphorylation at S45. Taken together, these findings suggest that ARMC10 is a key AMPK substrate that functions in mitochondrial fission.

The mitochondria represent the primary site of ATP production. Given that AMPK function is directly regulated by AMP:ATP or ADP:ATP ratio, and AMPK is the key kinase regulating energy homeostasis, it is not surprising that AMPK participates directly in the control of mitochondrial dynamics and functions. We speculate that ARMC10 directly facilitates mitochondrial fission and contributes to mitophagy.

ARMC10 is located at the outer membrane of the mitochondria. We validated the binding between ARMC10 and two mitochondria fission proteins, MFF and FIS1, which have been reported as receptors that recruit Drp1 to the mitochondrial surface[41], whereas Drp1 mediates membrane fission by wrapping around and constricting mitochondrial tubules[43,44]. We further demonstrated an association between ARMC10 and Drp1, indicating that ARMC10 may work together with MFF, FIS1, and Drp1 to mediate mitochondrial fission. The details of the underlying mechanisms remain to be elucidated.

This and previous studies indicate that AMPK has important roles in the regulation of mitochondrial dynamics and other processes that are critical for cellular response to energy stress. We are still at the early stages of understanding the detailed mechanisms underlying AMPK-dependent regulation of many cellular processes involved in energy consumption and generation. We expect that our unbiased global phosphoproteomic analysis of AMPK-dependent phosphorylation events will enable us to gain a better understanding of the AMPK-dependent signaling network.

## Methods

**Cell culture and transfection.** HEK293A, HEK293T and U2OS cells were purchased from ATCC (Manassas, VA) and maintained in Dulbecco modified Eagle medium supplemented with 10% fetal bovine serum, 1% penicillin and streptomycin at 37 °C in 5% $CO_2$ (v/v). AMPKα1/α2-DKO HEK293A-derived, ARMC10-KO HEK293A-derived, and U2OS-derived cells were generated by the CRISPR-Cas9 system. The KO cells were validated by Sanger sequencing and immunoblotting as shown in Supplementary Figures 1a, b, 4e, f. AMPKα1/α2-DKO U2OS-derived cells were a kind gift from Dr. Reuben Shaw (Salk Institute for Biological Studies).

For plasmid transfection, cells were seeded in 6-well plates. The next day, a mixture of 2 μg DNA, 10 μL of polyethylenimines and 200 μL Opti-MEM (ThermoFisher Scientific, Waltham, MA) was added into one well of each plate. After incubation for 16–24 h, cells were collected or treated as indicated.

**Plasmid and antibodies.** Constructs were generated via polymerase chain reaction (PCR) and subcloned into a pDONOR201 vector using Gateway Technology (Invitrogen, Carlsbad, CA) for use as the entry clones. As needed, the entry clones were subsequently recombined into Gateway-compatible destination vectors for the expression of N- or C-terminal-tagged fusion proteins. PCR-mediated site-directed mutagenesis was used to generate serial point mutations as indicated. The human codon-optimized Cas9 construct and target gRNA expression construct were obtained from Addgene.

Anti-ARMC10 S45 phospho-specific antibody (1:500) was raised against KLH-conjugated phospho-peptide GIRSSK(phospho-S)AED. Antisera were precleared with non-phosphopeptide with the same sequence and then affinity-purified by using the AminoLink Plus immobilization and purification kit (Pierce, ThermoFisher Scientific). Anti-AMPKα1 (2795 S), anti-AMPKα2 (2757 S), anti-AMPKβ (5831 S), anti-ACC1 (3676 S), anti-phospho-AMPK T172 (2535 S), anti-phospho-ULK1 S555 (5869 S), anti-phospho-ACC1 S79 (11818), and anti-COX IV (4850 S) were purchased from Cell Signaling Technology (Beverly, MA) and used at 1:1000 dilution. Anti-phospho-Ser antibody was purchased from Santa Cruz Biotechnology (Santa Cruz, CA) and used at 1:200 dilution. Anti-α-tubulin (T6199-200UL) and anti-Flag (M2) (F3165-5MG) monoclonal antibodies were purchased from Sigma-Aldrich (St Louis, MO) and used at 1:5000 dilution. Anti-ARMC10 (20506-1-AP) was purchased from Protentech (Chicago, IL) and used at 1:500 dilution. Anti-TOM20 (sc-11415) was purchased from Santa Cruz Biotechnology and used at 1:500 dilution in immunofluorescence analysis.

**CRISPR-Cas9-mediated gene knockout.** gRNA sequences were introduced into an expression construct using site-directed mutagenesis via PCR. The gRNA sequences for the indicated genes were as follows:
   AMPKα1, 5′-TCCTGTTACAGATTGTATGC AGG-3′;
   AMPKα2, 5′-ACGTTATTTAAGAAGATCCG AGG-3′;
   ARMC10, 5′-TTACCTGCTGGAGTCAACGG AGG-3′.
Constructs encoding Cas9 and gRNA were cotransfected with pCDNA3-enhanced green fluorescent protein (GFP) into HEK293A or U2OS cells by using polyethylenimine transfection. Cells were sorted based on their GFP signals, and KO clones were screened by Western blotting with corresponding antibodies and confirmed by sequencing.

**SILAC and protein digestion.** For SILAC experiments, AMPKα1/α2-DKO HEK293A and parental HEK293A cells were labeled via passaging for up to eight cell divisions in Dulbecco modified Eagle medium containing L-arginine (Arg 0) and L-lysine (Lys 0) ("light") or L-arginine-U-$^{13}C_6^{15}N_4$ (Arg 10) and L-lysine-U-$^{13}C_6^{15}N_2$ (Lys 8) ("heavy"). The medium was supplemented with 10% dialyzed fetal bovine serum. Labeling efficiency was tested periodically using MS. Cells in light or heavy medium were treated or not treated with 100 μM A769662 for 24 h. Cells were collected and mixed immediately.

Cells were solubilized in GdmCl lysis buffer (6 M GdmCl, 100 mM Tris, pH 8.5, 10 mM dithiothreitol). Cells were then subjected to sonication and centrifugation for 30 min at $4000 \times g$ at 4 °C. Supernatants were placed in a clean tube and reduced in 5 mM dithiothreitol for 1 hour at 56 °C followed by alkylation with 20 mM iodoacetamide for 45 min at room temperature in the dark. Volumes (4×) of −20 °C acetone were added to precipitate the protein overnight. Precipitated protein was collected via centrifugation for 20 min at $2000 \times g$ at 4 °C. Pellets were washed twice with −20 °C 80% acetone and air-dried. Pellets were resuspended in 8 M urea in 50 mM $NH_4HCO_3$ buffer with sonication at 4 °C for 5 min. After proteins were quantified by using a Bradford assay, 2 mg of each protein was placed in a new tube, and the urea was diluted with 50 mM $NH_4HCO_3$ buffer to reduce the concentration of urea to <1 M. The proteins were then digested in trypsin (1:100) for 18 h at 37 °C ($1000 \times g$).

**Phosphopeptide enrichment.** The phosphopeptides were enriched according to a published protocol[45]. After digestion, 150 μL of 3.2 M KCl, 55 μL of 150 mM $KH_2PO_4$, 800 μL of acetonitrile (ACN), and 95 μL of trifluoroacetic acid were added to the digested peptides. Peptides were mixed at room temperature for 1 min at $2000 \times g$, cleared via centrifugation and transferred to a clean tube.

$TiO_2$ beads were added to peptides at a bead:protein ratio of 10:1, suspended in 80% ACN/6% trifluoroacetic acid and incubated at 40 °C for 5 min at $2000 \times g$. The beads were then pelleted via centrifugation for 1 min at $3500 \times g$, and the supernatant (containing non-phosphopeptides) was aspirated and discarded. The beads were suspended in a wash buffer (60% ACN and 1% trifluoroacetic acid), transferred to a clean tube and washed four more times with 1 mL of wash buffer

each time. After the final wash, the beads were suspended in 100 μL of a transfer buffer (80% ACN and 0.5% acetic acid), transferred onto the top of a C18 StageTip (ThermoFisher Scientific) and subjected to centrifugation for 5 min at $500 \times g$ or until no liquid remained on the StageTip. Bound phosphopeptides were eluted twice with 30 μL of an elution buffer (40% ACN and 15% NH$_4$OH; high-performance liquid chromatography grade) and collected via centrifugation into clean PCR tubes. Samples of the phosphopeptides were concentrated in a SpeedVac for 15 min.

**Mass spectrometry analysis.** Phosphopeptide samples were fractionated followed a described protocol[46]. Vacuum-dried peptides were dissolved in pH 10 buffer (10 mM ammonium bicarbonate, pH 10, adjusted by NH$_4$OH) and subjected to pH 10 C18 reverse-phase column chromatography. Peptides were eluted with a step gradient of 150 μL each of 6, 9, 12, 15, 18, 21, 25, 30, and 35% ACN (pH 10), pooled into six pools and vacuum-dried for nano-high-performance liquid chromatography-tandem MS.

Peptides were loaded onto a 10-cm column with a 150-μM inner diameter, packed in-house with 1.9 μM C18, and subjected to nano-high-performance liquid chromatography-tandem MS analysis by using a nano-LC 1000 coupled with an Orbitrap Elite mass spectrometer (ThermoFisher Scientific). The peptides were separated with a 75-min discontinuous gradient of 2–24%, 4–24%, or 8–26% ACN/0.1% formic acid at a flow rate of 800 nL/minute. The mass spectrometer was set to data-dependent mode, the precursor MS spectrum was scanned at 375–1300 m/z with 240k resolution at 400 m/z ($2 \times 10^6$ AGC target), and the 25 strongest ions were fragmented via collision-induced dissociation with 35 normalized collision energy and 1 m/z isolation width and detected by using an ion trap with 30 s of dynamic exclusion time, $1 \times 10^4$ AGC target, and 100 ms of maximum injection time.

**Mass spectrometry data analysis.** Raw MS data were processed by the Max-Quant software program (version 1.5.2.8; Max Planck Institute of Biochemistry, Martinsreid, Germany) with a false-discovery rate <0.01 at the protein, peptide, and modification level using default settings with the following minor changes: multiplicity was set to 2 with light and heavy labeling; oxidation of methionine, acetylation of N-term, and Phospho (STY) were selected as variable modifications; carbamidomethyl(C) was selected as a fixed modification; and the minimum peptide length was 7 amino acids. Proteins and peptides were identified by using a target-decoy approach with a reversed database. The Human UniProt FASTA database (October 2015) containing 70,097 entries was searched. Peptides and proteins were quantified by using MaxQuant. Bioinformatics analysis of the proteomics results was done with Microsoft Excel and R statistical computing software. Significance was assessed with $t$-tests. The differentially expressed phosphopeptides were subsequently filtered for median fold-change >1.5 and $p$ value < 0.05 ($t$-test). A site localization probability of 0.7 was used as the cut-off for localization of phosphorylation sites. Conserved motif analysis figures for the changed phosphosites were generated using WebLogo[47]. Function annotation was done by Ingenuity Pathway Analysis software (QIAGEN, Redwood City, CA) and reference mining.

**In vitro kinase assay.** Recombinant WT ARMC10 and the ARMC10 S45A mutant were expressed in bacteria and purified with use of a maltose-binding protein tag. Both recombinant proteins were mixed with active recombinant AMPK (14–840; Merck Millipore, Burlington, MA) and γ-$^{32}$P-ATP (NEG002A100UC; PerkinElmer, Waltham, MA) in a kinase buffer (25 mM Tris, pH 7.5, 5 mM β-glycerophosphate, 2 mM dithiothreitol, 0.1 mM Na$_3$VO$_4$, 10 mM MgCl$_2$). These samples were incubated at 30 °C for 15 min. The reactions were stopped via boiling, and proteins were separated using sodium dodecyl sulfate-polyacrylamide gel electrophoresis. The gel was dried, and it was imaged by using the PhosphorImager software program[10].

**Pulldown and immunoprecipitation.** For pulldown and immunoprecipitation assays, $1 \times 10^7$ cells were subjected to lysis with NETN buffer on ice for 30 min. The lysates were incubated with 20 μL of conjugated streptavidin beads or S-beads (for SFB-tagged pulldown) or with 20 μL of protein A/G agarose together with indicated antibodies for 2 h at 4 °C. Beads were washed three times with NETN buffer and boiled in 2× Laemmli buffer before Western blot analysis[48]. Uncropped scans of all Western blots are included in Supplementary Figure 7.

**Immunofluorescent staining.** Cells cultured on coverslips were fixed in 3% paraformaldehyde for 10 min at room temperature and then extracted with a 0.5% Triton X-100 solution for 5 min. After blocking with a Tris-buffered saline/Tween 20 solution containing 1% bovine serum albumin, cells were incubated with the indicated primary antibodies for 1 hour at room temperature. After that, cells were washed and incubated with fluorescein isothiocyanate or rhodamine-conjugated second primary antibodies (1:3000 dilution; Jackson ImmunoResearch Laboratories, West Grove, PA) for 1 hour. Cells were counterstained with 100 ng/mL 4′-6-diamidino-2-phenylindole for 2 min to visualize nuclear DNA. The coverslips were mounted onto glass slides with anti-fade solution and visualized using an Eclipse

E800 fluorescent microscope with a Plan Fluor 60× oil objective lens (numerical aperture, 1.30; Nikon Instruments, Melville, NY).

Quantification of mitochondrial morphology followed the strategy reported elsewhere[11]. In brief, the quantification was scored from more than 150 cells with three replicates, and it was done by a person who was blinded to genotype and treatment of the samples. Morphology was scored as follows: "fragmented", the majority of mitochondria were spherical; "short", the majority of mitochondria were less than ~10 μm; "long", the majority of mitochondria were more than ~10 μm; and "fused", highly interconnected mitochondria with fewer than 4–5 free ends.

**BioID pulldown.** BioID2 (Addgene) cDNA was amplified with ARMC10 at the N-terminus and inserted into a pDonor vector. After a subsequent reaction, BioID2-bait pDEST constructs were obtained (Gateway cloning protocols). The constructs were transfected into HEK293T cells by using polyethylenimine. HEK293T cell lines with stable expression of tagged proteins were validated with Western blotting after puromycin selection. For the BioID pulldown experiment, $4 \times 10^7$ cells were collected after incubation with 50 μM biotin for 16 h. Cells were subjected to lysis and cleared via centrifugation. The supernatants were incubated with 100 μl of streptavidin agarose beads overnight. After extensive washing, the samples were eluted by boiling with 80 μl of 2× Laemmli buffer, then separated by SDS-PAGE and visualized by Coomassie Blue staining.

The gel band containing the entire sample was excised into small pieces and destained completely before digested with trypsin at 37 °C overnight. The peptides were extracted with acetonitrile and vacuum dried. The mass spectrometry analysis setting is similar to the method part of mass spectrometry analysis with a few changes. The samples were subjected to nanoscale liquid chromatography (EASY-nLC 1000 liquid chromatography system, ThermoFisher) coupled to tandem mass spectrometry (Q Exactive Plus, ThermoFisher) analysis. The peptides were separated with a 75-min discontinuous gradient of 4–24% ACN/0.1% formic acid at a flow rate of 800 nl/minute. The MS raw data were searched in Proteome Discoverer 1.4 (ThermoFisher) with Mascot algorithm 2.4 (Matrix Science). The identified peptides and proteins were filtered by false discovery rate <1%.

Six repeats of the BioID2-tagged ARMC10 experiment were performed for the bait, and eight repeats were performed for the controls, which included BioID2 tag fused with mitochondria localization sequence (two repeats) or fused with a nonrelated bait protein, SLX4IP (six repeats). To identify the significantly enriched proteins, we compared the identification list from BioID2-tagged ARMC10 to the list from control groups. The results for samples and controls were compared via $t$-tests.

**Statistical analysis.** Each experiment was repeated two or more times unless otherwise noted. Differences between groups were analyzed by using the Student's $t$-test or two-way analysis of variance with the Tukey multiple comparisons test. $P$ values < 0.05 were considered statistically significant.

## Data availability

The mass spectrometry proteomics data have been deposited to the ProteomeXchange Consortium via the PRIDE partner repository with the dataset identifier PXD011696. All relevant data not presented in the main figures or Supplementary Data are available from the authors.

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

## Acknowledgements

We thank Drs. Yi Wang, Yin Ye, My Kim Tran, and Antrix Jain for their kind help, and Drs. Sebastien Herzig and Reuben Shaw for sharing unpublished observations. We also thank the Department of Scientific Publications at The University of Texas MD Anderson Cancer Center for editing the manuscript. This work was supported in part by startup funds from MD Anderson Cancer Center and Era of Hope Scholar Research Award W81XWH-09-1-0409 from the U.S. Department of Defense to J.C.; J.C. also received support from the Pamela and Wayne Garrison Distinguished Chair in Cancer Research. This project was supported in part by the National Institutes of Health through Cancer Center Support Grant P30CA016672.

## Author contributions

Z.C., C.L., J.Q., and J.C. conceived the project. Z.C., C.L., C.W., N.L., M.S., M.T., H.Z., J. M.C., and S.Y.J. performed the experiments. Z.C. and J.C. wrote the manuscript with input from all authors.

## Additional information

**Competing interests:** The authors declare no competing interests.

