## [Peer Review File · Nature Communications]

Reviewers' comments:

Reviewer #1 (Remarks to the Author):

This manuscript describes an approach to identify new AMPK targets using SILAC methodology in both wild type and AMPK knockout cells. From this screen they identify ARMC10 (which they call MFIN) as a novel target for AMPK. They go on to focus on this protein and present evidence that it is involved in activation of mitochondrial fission in response to AMPK activation. While this would be an interesting conclusion if confirmed, there are many problems that need addressing, as detailed below. A major problem with reviewing this manuscript was the fact that there was often insufficient information provided in Figure legends or the Methods section to allow the reader to work out exactly what had been done.

SPECIFIC POINTS:

1. The authors initially use the term MFIN to describe the protein on which they focus their attention. However, I could not find this name for a gene or protein in the NCBI, UNIPROT or HUGO databases. If MFIN is a new gene/protein name, the authors should state where this comes from. On line 76 they also use the term ARMC10 (Armadillo repeat-containing protein 10), which does appear in the databases and contains the phosphorylation site sequence that they identified. However, on all subsequent mentions in the manuscript it is referred to as AMRC10, which appears to be incorrect. These discrepancies need to be sorted out. While it is OK to mention alternate names at first mention, they should subsequently stick to a single name for consistency.

2. Line 27: "When the ATP level is lower than the AMP or ADP level, AMPK is activated" – this is not accurate – AMPK is activated by increases in the AMP:ATP and ADP:ATP ratios, but it would be very unlikely that the level of AMP or ADP would ever be higher than that of ATP, except perhaps in dying cells.

3. Lines 29-30: I do not understand what they mean by "prevents the use of intermediates in glycolysis" – please clarify.

4. Lines 57-58: this sentence implies that the use of AMPK knockout cells in the identification of downstream targets for AMPK is new, but in fact they were used in almost all of the studies referred to earlier, the only difference being that the CRISPR method was not used to create the knockouts. However, CRISPR is now a well-established method that is not truly novel in itself. This rather misleading claim for novel methodology is also made elsewhere, e.g. in lines 311-313 in the Discussion.

5. Fig. 1a: the lanes in the Western blot need labelling.

6. Fig. 1c/d: I would like to hear their justification for the use of 24 hr incubation with A769662 for target identification. Fig. 1c shows that the phosphorylation of ACC is maximal within 1 hour. It is true that phosphorylation of AMPK itself takes longer, but this is because A769662 is primarily an allosteric activator of AMPK and its effects on Thr172 phosphorylation are quite modest compared with other activators such as AICAR, at least at short time points (see e.g. Fig. 5 in Goransson et al (2007) J Biol Chem 282:32549). Use of a 24 hr incubation time is liable to increase the number of secondary events that are not directly caused by AMPK activation.

7. Fig. 1c: what concentration of A-769662 was used?

8. Fig. 1d: what incubation time and concentration of A-769662 – although this information is in the Methods section, it would be very helpful if it was also in the legend.

9. Supp. Fig. 1: the Figure (for which little background information is supplied) seems to imply that HEK-293 and U2OS cells contain only two alleles, but according to the CanSAR website these

cell lines are hypotriploid and hypertriploid respectively, so that there may be more than two alleles.

10. Fig. 2c: they mention in the Discussion that Ingenuity software was used for this analysis, but that information should be in the legend and Methods section. Also, it would be very useful to have a table where the proteins included are listed, along with their functional annotations.

11. Fig. 2d: why was LIPE not included in this list? Note also that the site on CTNNB1 does not really fit the AMPK consensus motif. Also, please check the residue numberings in this list (and in Supp. Fig. 2) carefully, as they do not always appear to correspond to the numbering in the current UNIPROT entry for that protein (e.g. NUMA1, REEP1). Note that MaxQuant sometimes assigns residue numbering using older database entries. Also, which paper previously identified S74 on MAST2? Although MAST2 is identified in Ducommun et al, they did not identify the actual site.

12. Supp. Fig. 2: what program was used to search the hits for the AMPK motif, and what flexibility in the motif was allowed? Some of the sites do not appear to be good fits for the AMPK consensus motif.

13. Lines 132-133: to avoid confusion, please give individual references to each target protein rather than grouping the references.

14. Lines 143-146: is the site they identify on KLC4 equivalent to the S539 or the S575 site on KLC2?

15. Fig. 3a: please explain in the legend what MFIN (S45)_1 and MFIN (S45)_2 refer to.

16. Fig. 3b: there does seem to be less signal with the P-Ser antibody for MFIN, KLC4 and REEP2, and with the p-AMPK substrate motif antibody for MFF, in the AMPK KO cells. However, the FLAG blots in the lower panel are over-exposed, and it is possible that in some cases there is also less protein in the AMPK KO cells.

17. Fig. 3c (also Suppl. Fig. 3a) : what does the color coding mean?

18. Fig. 3d: to enable the reader to understand what had been done, much more detail about the tags utilized, and the immunoprecipitation protocol used, needs to be provided in the Figure legend and/or Materials & Methods.

19. Fig. 3e: the effect of A769662 on phosphorylation of S45 is quite modest – it would be helpful to also have a pACC blot here as a positive control, to see how this compared.

20. Fig. 3f: the alignment of the ³²P autoradiogram with the blots seems to be faulty – it appears from this Figure that MFIN is phosphorylated better in the absence of AMPK! There is also a (less serious) problem with Fig. 3g, where it is probably only the "+/-" labels that are poorly aligned. Also, in the legends it states that "both isoforms of MFIN were phosphorylated by the AMPK activator A769662", but A769662 is not an agonist, not a protein kinase!

21. Supp. Fig. 3b: this Figure is not referred to in the text, and it is not clear what it adds.

22. Supp. Fig. 3c: it seems strange that the phospho-AMPK substrate motif antibody seems to detect many proteins in the lysate, but detects none of them after immunoprecipitation. Incidentally, it is presumed that some of the bands in these blots represent the heavy and light chains of the immunoglobulin used – it would have been helpful if these were labelled. Finally, is this Figure worth including at all?

23. Lines 193-195 and Figs. 4b: it appears that only the smaller isoform of MFIN is phosphorylated in intact cells – is that correct?

24. Lines 212-215 and Fig. 4c/d: they state that “we observed drastic changes in mitochondrial morphology only in cells with WT MFIN overexpression (Fig. 4c)”. While there does seem to be a difference in mitochondrial morphology between over-expression of WT MFIN and the S45A mutant, they do not analyse control (untransfected) cells in Figs. 4c or 4d, so you cannot really conclude that there is a drastic change on over-expression of the WT. Also, in Fig. 4e there does not seem to be much change in mitochondrial morphology on over-expression of the wild type.

25. Fig. 4d/f: here they have divided mitochondria up into four morphological categories, but there are no details of how this was done (e.g. by manual inspection, or using software?). Also, when they do statistical analysis to compare columns, which categories of mitochondrial morphology (e.g. long, short, fused or fragmented) are they comparing? More explanation is needed.

26. Figs. 4a/b/g: these panels really seem to belong more in Fig. 3 (Validation of AMPK substrates) rather than Fig. 4 (Function assay of MFIN and MFIN phosphorylation at S45).

27. Fig. 4a: the difference in MFIN phosphorylation between lanes 1 and 2 is modest, and the FLAG and tubulin blots are over-exposed so it is difficult to say that the amount of protein loaded is the same.

28. Fig. 4e/f and lines 235-238: they state that cells expressing the S45E mutant had increased mitochondrial fission and fragmentation, but according to Fig. 3f there does not appear to be any significant increase in the proportion of fragmented mitochondria.

29. Fig. 4g: because there is quite a high basal MFIN phosphorylation in the WT cells, it would be good to see a pACC blot as a positive control.

30. Fig. 4c: if they were to provide unmerged images, this would allow the reader to more easily compare mitochondrial morphology in transfected and untransfected cells.

31. Supp. Fig. 4c: there is no quantification of these results and no indication of how many times the experiment has been performed.

32. Supp. Fig. 4d: there is no explanation provided of how this Table was arrived at, or what the numbers mean.

33. Fig. 5a/b: to my eyes, mitochondrial morphology looks different in Fig. 5a in the WT and KO after DMSO treatment, whereas in the quantification in Fig. 5b they look almost identical.

34. Lines 252-256: this sentence equates glucose starvation with energy stress, whereas it has been shown that, depending on the cell type, glucose starvation is not always associated with changes in adenine nucleotide levels, although AMPK is still activated [Zhang et al (2017) Nature 548: 112]. I suggest replacing “energy stress” simply with “glucose starvation”.

35. Fig. 6a: this is an example of a Figure where more documentation of the experiment is required.

36. Figs. 6a/b: why does MFF appear as one band when pulled down with SFB but at least three bands when pulled down with anti-myc?

37. Fig. 6e: are there arrows missing from this model? More explanation is required.

38. Fig. 6d: why does myc-tagged MFIN and MFF appear to run at 100 kDa, whereas in 6b and 6c

it runs at around 40 kDa?

39. The Discussion section is rather repetitive of the Results section and could be condensed – it should discuss the results rather than just restating them.

40. Lines 367-369: the authors claim that over-expression of MFIN in U2OS cells led to mitochondrial fragmentation, but I could not find any Figures to show this.

42. Lines 377-379: it would be more appropriate to say that AMPK is regulated by AMP and ATP, rather than ATP alone.

Reviewer #2 (Remarks to the Author):

Ncomms-18-15823 Global phosphoproteomic analysis reveals MFIN as an AMPK substrate that regulates mitochondrial dynamics

AMPK is a master regulator of cellular energy homeostasis. It is a heterotrimeric complex with one catalytic subunit (alpha) and two regulatory subunits (beta and gamma). The goal of this work is to identify AMPK substrates by comparing global quantitative proteomic phospho-peptides in wild type and AMPK knock out cells following activation of AMPK by a small molecule A769662, which prevents dephosphorylation of Thr172 on the alpha subunit.

Of 32 AMPK phosphopeptides identified in the analysis, twenty were novel. One was chosen for a more extensive analysis: AMRC10 (MFIN). This protein is localized to the mitochondrial outer membrane and authors argue that it plays a AMPK-regulated role in mitochondrial division. The validation of AMRC20/MFIN as an AMPK substrate is strong; however, the data quantifying mitochondrial morphology and the role of this pathway in cell death does not support the conclusions at this time.

Specific comments:

In Figure 3b, five potential protein targets of AMPK are overexpressed in wild type or AMPK DKO HEK cells treated with A769662, immunoprecipitated and subject to western blot analysis with a phospho-Serine or P-AMPK substrate antibody.

The P-AMPK antibody only recognized Mff, a previously reported substrate of AMPK. The P-Ser antibody has signal for MFIN, KLC4 and REEP2. Therefore, REEP1 was not detected by either antibody.

In all cases, there is significant signal in the double knock out cells, which should lack AMPK activity. Authors should comment on the source of the phosphorylated protein in these cells. Alternatively, an alanine mutant could be generated for each predicted site and expressed as in Figure 3e. As presented, the data are not a compelling argument that the new targets are bona fide, and could be removed or moved to supplement.

In Figure 3d, interaction of overexpressed MFIN is immunoprecipitated from cells and to test for a physical interaction with AMPK-alpha, CAMKK-beta and LKB1. Authors should discuss how CAMKK-beta and LKB1 were chosen as negative controls. For example, do these often share substrates with AMPK?

Phosphoblock mutants and in vitro kinase activity assays suggest that MFIN/AMRC10 is a bona fide AMPK substrate. In Figure 3f, the individual lanes in the three panels of the gel are not aligned, making it difficult to interpret the data and raising the possibility that these are not from the same experiment or gel.

Both the short and long isoform of the protein is included in Figures 3f, 3g and 3h, but this is not described clearly in the text.

Overexpression of N-terminally tagged MFIN/AMRC10 in HEK293T cells appears to cause mitochondrial fragmentation, while the S45A mutant does not.

Vector controls are lacking from this experiment and make it impossible to determine the degree to which the mitochondrial morphology is altered in cells expressing MFIN/AMRC10.

The metric for scoring mitochondrial morphology must be included in the material and methods. For example, how were long and fused mitochondria distinguished? How many cells were counted? How many times was the experiment repeated? Were the samples blinded?

To determine if the phosphorylation state at Ser-45 alters the activity of MFIN/AMRC10, the phosphomimetic mutant S45E is expressed in wild type and AMPK DKO cells. Activation of AMPK activity in wild type cells induces mitochondrial fragmentation, but not in the DKO cells. Regardless of AMPK activation, cells expressed MFIN/AMRC10 S45E are scored as having a higher proportion of cells with fragmented mitochondria.

The text is not consistent with the metric used on the graph, where the increased population is referred to as "short".

This analysis also lacks a vector control for reference.

The immunostaining pattern is odd in these experiments. Tom20, a mitochondrial outer membrane protein, is labeled with green and MFIN/AMRC10 is labeled with red. In contrast to the wild type and S45A variants, the S45E variant does not appear to completely co-localized with Tom20.

Specifically, the perinuclear mitochondria are labeled only with Tom20 and the peripheral mitochondrial with MFIN/AMRC10 S45E. It looks like the Tom20 signal is mitochondrial aggregates but the MFIN/AMRC10 are not. This raises the possibility that the tag or the mutation is altering the function of MFIN/AMRC10 and may not represent an accurate assessment of protein function.

CRIPSR-Cas9 knock out of MFIN/AMRC10 are generated in U2OS cells. Compared to controls, the knock out cells have extended/more connected mitochondria. In contrast to the wild type control, the mitochondria in the knock out cells do not fragment with activation of AMPK following A769662 treatment. These data argue that MFIN/AMRC10 plays a role in determining mitochondrial structure.

The N-terminally tagged variant utilized in previous experiments should be expressed in the knock out cells to assess function, which could be altered by the tag.

To further substantiate that MFIN/AMRC10 activity is regulated, the S45A and S45E mutants should be expressed in these cells and mitochondrial morphology scored with and without A769662 treatment.

Mitochondrial dynamics influence susceptibility to cell death. Cell survival was assessed following energy stress (glucose starvation) to determine if MFIN/AMRC10 plays a role in this pathway. The Figure Legend suggests that all cell lines are HEK293T; was MFIN/AMRC10 also knocked out in these cells?

Assuming that the MFIN/AMRC10 knock out cells are U2Os, a wild type U2OS cell line is required in this analysis, to demonstrate that this cell type has a comparable response to wild type HEK293T cells.

Minor Comments:

Figure 1A lacks labels for each lane.

Figure 1C legend should state that these are wild type cells.

Authors switch between AMRC10 and MFIN throughout the manuscript. MFIN is very close to MFN, which is a mitochondrial fusion factor, and could be a confusing acronym to adopt.

Lines 259 – 273 discuss a connection with mitophagy, which has been previously correlated with

mitochondrial fragmentation. Without additional experiments, this data seems preliminary and difficult to draw a conclusion from.

Reviewer #3 (Remarks to the Author):

In the manuscript titled 'Global Phosphoproteomic Analysis Reveals MFIN as an AMPK Substrate that Regulates Mitochondrial Dynamics', the authors discuss the use of a double knockout cell line to identify phosphosites directly or indirectly regulated by AMPK, followed by the validation of identified phosphosites and investigation of the MFIN interactome. Although the work presented is of interest it still requires a major rewrite and additional analysis, mainly related to the clarification of the experimental setup and proteomics data analysis. Please find specific comments below.

Main comments

1. The authors could benefit from a more detailed description of the reasoning behind their experiments. Currently, it is not clear why certain experiments were performed or why certain protein candidates were chosen for further experiments. For example, the importance of Fig 1C is not discussed in the main text nor figure legend, whereas it is probably the basis for the decision to stimulate cells with A769662 for 24 hours (mentioned much later in the text).
2. The authors do not clearly specify their bioinformatics workflow. At present it is unclear whether they calculate ratios between phosphosites or identified phosphopeptides, how many unique and high-confidence phosphosites were identified (as they only indicate the number of phosphopeptides) and the cutoffs used for filtering the data are not supplied. The authors should specify cutoffs used for localization probabilities, identification in biological replicates, and so on. The use of a median in the low number of replicates is questionable.
3. When investigating MFIN (Fig 3F, 4B, 4G) it seems that the MFIN-343-S45A mutant is still phosphorylated by AMPK in vitro (Fig. 3F). When assessing phosphorylation in cell lines, it appears only one of the two isoforms of MFIN is actually phosphorylated (Fig 4B and 4G), as indicated by a double MFIN band versus a single Phos-MFIN band. Yet both isoforms were reported as phosphorylated based on the phosphoproteomics study. This discrepancy is confusing and should be discussed.
4. On page 14, line 277 and onwards, the authors discuss a BioID experiment where they investigate the MFIN interactome. The setup of this experiment is not clear, please specify control bait protein, experimental conditions and elaborate on data analysis leading to Supplemental Figure 4D. Also, in Supplemental Figure 4D, the meaning of numbers listed under the columns "BioID2_MFIN" and "control" is unclear.

General comments

1. The discussion is currently resembling a summary of the data, we feel the authors could make better use of the discussion by discussing the major drawback of their technique, namely the risk of comparing a dysregulated/stressed cell line with a WT cell line and the inability to distinguish whether differences in phosphosites are due to the absence of AMPK or a general stress response in the cell.
2. On page 4 line 70, the authors state that "51 phosphosites were found to have lower levels in AMPK WT cells than those in AMPK DKO cells, suggesting that these are likely phosphorylation events negatively and probably indirectly regulated by AMPK", however, since the authors generated a permanent DKO of a major kinase complex, these sites may very well reflect some of the cell's compensating mechanisms. This possibility should be addressed by the authors. This possibility is reflected in supplemental figure 1A, where the knockout of AMPKa1 clearly resulted in an upregulation of AMPKa2 and vice versa.
3. On page 4, line 72, the authors use the "known conserved AMPK phosphorylation motifs" to distinguish directly and indirectly regulated AMPK substrates, however, the authors do not specify what they define as the conserved motif. Moreover, filtering on known motifs could hamper the identification of new phosphosites or substrates, especially when the known substrates do not share a strict consensus sequence (e.g. for AMPK, the phosphosite database revealed 65 known substrate sequences, without very strict amino acid conservation between them).

4. On page 9, line 181: The authors indicate that the MFIN S45A mutant was barely detectable with phospho-S antibody. It is unclear whether these HEK cells still express WT MFIN. If not, the presence of some phospho-S signal on MFIN S45A implies that there is a second, less abundant phosphorylation, which could mean the interaction between MFIN and AMPK is based on a different site altogether. In addition, the authors do not state which of the isoforms of MFIN was transiently expressed (or if both were expressed simultaneously). In Figure 3D, the authors only show AMPKa1, while AMPKa2 would also be informative.
5. In figure 2A, the significance of the colours is not specified. Also, MFIN is indicated with a larger diameter dot, the meaning of which is also not specified. How authors move from identified phosphopeptides to phosphosites is not clear.
6. Figure 2C appears to be a very elegant representation of enriched GO terms and localization, however, it is hardly mentioned in the main text and the meaning of the number of dots per GO term is not specified. The authors should consider discussing the figure in a bit more detail. Also, the legend may be simplified by referring back to the groups indicated.

Minor comments

1. The authors could benefit from a general spell/style check, to ensure they use consistent naming (e.g. Crispr/Cas9, the use of AMRC10 and MFIN interchangeably, the use of KO and DKO interchangeably, the use of phospho Ser and phosphor Ser antibody, all of which create confusion).
2. Figure 1D is an important control to assess the influence of labeling, however, as it currently does not add to the story as described by the authors, it may be better situated in the supplementary figures.
3. In supplementary table 1, the meaning of "positions within proteins" (column 2) is unclear. Are these the identified phosphosites? Do they all have confident phosphosite assignment? Please clarify.
4. In supplementary table 2, quite a few ratios of 0 are reported (e.g. for identifier P10071), which are left empty in supplementary table 1, indicating the corresponding phosphopeptides were not detected. Listing a zero in a log₂ ratio indicates the measured values are identical, which is not the case.
5. On page 6, line 114, the authors state: "51 phosphosites with lower levels in WT cells than those in AMPK DKO cells". At present it is not clear whether the same cutoff was used as for upregulated phosphopeptides or if a different cutoff was used.
6. The authors classify the 109 differentially regulated phosphosites in three groups, however, only group 1 is listed in Supplementary figure 2. It might be interesting to also show groups 2 and 3 and whether the downregulated phosphosites also comply with the AMPK motif (indicating a more complex regulatory system).
7. In figure 2D, the authors list the known AMPK substrates that they have identified in group 1. At present, the references to the corresponding articles are missing from this table. The rationale for distinguishing low and high throughput studies is not clear and hence confusing. Does this mean that the sites indicated as low throughput have been biologically validated while the sites classified as high throughput were not?
8. Currently, the authors do not emphasize the 20 novel phosphosites that they report. In order to find these, the reader needs to use supplementary figure 2 and manually substract the 12 previously reported sites listed in Figure 2D. Why not clearly show the 20 potentially new sites that the authors have identified in this study?
9. On page 8, line 142-144, the authors speculate around the role of AMPK phosphorylation of KLC4. This speculation would be supported by some additional information regarding sequence conservation between the two proteins or localization of the phosphosites on highly conserved domains. However, the authors do not comment on the similarity between the proteins, other than them sharing name and family. Interestingly, the authors speculate a similar biological function for KLC4 in their U2OS/Hek293 cells, however, axon-dendrite polarization might not be expected to occur in these cell lines.
10. On page 8, line 145, the authors chose 5 sites in group 1 for further analysis. Please specify which sites were chosen and why.
11. In figure 3a, the authors show the log₂ ratio between WT and DKO cells for 5 proteins across

the four replicates. This figure may benefit from showing average values and standard deviations instead of individual values for each replicate. At present, the figure emphasizes the variation between replicates. In addition, the difference between MFIN_S45_1 and MFIN_S45_2 is not clear here, and not explained in the text until page 9 line 168.

12. On page 8, line 155, the authors discuss the validation of five selected sites by western blot. They state that detection of MFF and REEP1 is hampered "probably because of the low affinity of this phospho specific antibody for western blotting". If this antibody is in fact low affinity, this should be visible in all conditions. More likely, the authors are experiencing a problem of specificity and/or sensitivity.

13. In figure 3B, the indications of AMPKa1/a2 presence overlap, since all conditions are +/+ or -/-, perhaps a single indicator would suffice. In addition, the anti-flag antibody seems to indicate some truncations, which appear dependent on the presence of AMPKa1/a2 and are not discussed at all. In addition, it is unclear whether the P-AMPK substrate motif antibody binds the phosphorylated motif or the non-phosphorylated motif. More generally, it is unclear for this figure how the experiment was performed. Was the pulldown performed with P-Ser and P-AMPK antibodies or with the tagged proteins themselves? Please clarify.

14. Supplementary figure 3 shows three panels, A-C, which are only explained in the legend. For supplementary figure 3A, the relevance of the colours is unclear. In 3B, it is unclear which extracts are meant and why this figure is in the manuscript in addition to figure 1B. For 3C, the meaning of the red boxes is unclear as well as the identity of many of the bands shown. Please include molecular weight markers. Also, it seems the figure serves to strengthen the point the authors make on p. 8 line 155, however, the figure is not referred to in that section.

15. In Figure 3G, it is unclear why mock transfection in presence of AMPK complex does not show any MFIN or phospho MFIN.

16. In Figure 4C and 4E, scale bars are missing. In Figure 4D, 4F and 5B, it is unclear how Fused/short/long/fragmented are defined.

17. On page 12 line 239, the authors mention MFIN KO U2OS cells. Based on Supplementary Figure 1B, this is actually a double KO or a conserved region was targeted to knock both isoforms out in a single step. However, this is not specified.

18. In supplemental Figure 4B, the indices 2 and 3 of the zooms have been switched. Also, scale bars are missing. It is unclear whether overexpression of MFIN induces mitochondrial aggregation or that MFIN aggregates on the mitochondria upon overexpression.

Point-by-point response to reviewers' comments:

Reviewer #1 (Remarks to the Author):

This manuscript describes an approach to identify new AMPK targets using SILAC methodology in both wild type and AMPK knockout cells. From this screen they identify ARMC10 (which they call MFIN) as a novel target for AMPK. They go on to focus on this protein and present evidence that it is involved in activation of mitochondrial fission in response to AMPK activation. While this would be an interesting conclusion if confirmed, there are many problems that need addressing, as detailed below. A major problem with reviewing this manuscript was the fact that there was often insufficient information provided in Figure legends or the Methods section to allow the reader to work out exactly what had been done.

Thank you for the nice summary, careful review, and great comments. As you will see below and also in the revised manuscript, we have now provided additional information in Figure legends, Methods, and main text to help the readers.

SPECIFIC POINTS:

1. The authors initially use the term MFIN to describe the protein on which they focus their attention. However, I could not find this name for a gene or protein in the NCBI, UNIPROT or HUGO databases. If MFIN is a new gene/protein name, the authors should state where this comes from. On line 76 they also use the term ARMC10 (Armadillo repeat-containing protein 10), which does appear in the databases and contains the phosphorylation site sequence that they identified. However, on all subsequent mentions in the manuscript it is referred to as AMRC10, which appears to be incorrect. These discrepancies need to be sorted out. While it is OK to mention alternate names at first mention, they should subsequently stick to a single name for consistency.

Thank you for your comment. MFIN is a new gene/protein name we used based on its function discovered in this study. This gene is also known as ARMC10 (armadillo repeat-containing protein 10) or SVH (splicing variant involved in hepatocarcinogenesis protein). The name ARMC10 comes from the fact that this protein contains armadillo repeat. As suggested by you and other reviewers, we decide to avoid any confusion in the literature and will use ARMC10 consistently in our revised manuscript.

2. Line 27: “When the ATP level is lower than the AMP or ADP level, AMPK is activated” – this is not accurate – AMPK is activated by increases in the AMP:ATP and ADP:ATP ratios, but it would be very unlikely that the level of AMP or ADP would ever be higher than that of ATP, except perhaps in dying cells.

Thank you for the comment. Yes, AMPK is activated when the ratios of AMP:ATP and ADP:ATP increase. We corrected this statement in our revised manuscript as following: “When the ratios of AMP:ATP and ADP:ATP increase, AMPK is activated...”

3. Lines 29-30: I do not understand what they mean by “prevents the use of intermediates in glycolysis” – please clarify.

The statement “prevents the use of intermediates in glycolysis” is intended to describe the situation in which activation of AMPK blocks the anabolic pathways to prevent the consumption of ATP and other intermediates. These intermediates such as pyruvate can be used to generate glucose.

To avoid any misunderstanding, we modified the sentence to “It switches on the glucose uptake and other catabolic pathways to generate ATP, while switches off the anabolic pathways to prevent the consumption of ATP, such as the conversion of glucose to glycogen”

4. Lines 57-58: this sentence implies that the use of AMPK knockout cells in the identification of downstream targets for AMPK is new, but in fact they were used in almost all of the studies referred to earlier, the only difference being that the CRISPR method was not used to create the knockouts. However, CRISPR is now a well-established method that is not truly novel in itself. This rather misleading claim for novel methodology is also made elsewhere, e.g. in lines 311-313 in the Discussion.

The reviewer is correct that knockout or knockdown is widely used in biologic studies and CRISPR is now an established method. As suggested, we removed any statement of “new” or “novel” in our revised manuscript, but only point out that we significantly reduced the background in our study with the use of AMPK catalytic subunit knockout cells. These catalytic knockout cells serve as an ideal control for phosphoproteomic studies and we will use this strategy in our future studies.

5. Fig. 1a: the lanes in the Western blot need labelling.

Thank you. We labeled the lanes in the revised Figure 1a.

6. Fig. 1c/d: I would like to hear their justification for the use of 24 hr incubation with A769662 for target identification. Fig. 1c shows that the phosphorylation of ACC is maximal within 1 hour. It is true that phosphorylation of AMPK itself takes longer, but this is because A769662 is primarily an allosteric activator of AMPK and its effects on Thr172 phosphorylation are quite modest compared with other activators such as AICAR, at least at short time points (see e.g. Fig. 5 in Goransson et al (2007) J Biol Chem 282:32549). Use of a 24 hr incubation time is liable to increase the number of secondary events that are not directly caused by AMPK activation.

Thank you for your comment. In this study, we wanted to achieve the highest activation of AMPK kinase activity. As reported, A769662 allosterically activates AMPK complex and also inhibits dephosphorylation of Thr-172 on the AMPK α subunit. Phosphorylation of Thr-172 in the activation loop of AMPK α is indicative of AMPK kinase activity. As shown in Figure 1c, Thr-172 phosphorylation increased with time and reached the highest level at the 24-hour time point. Similarly, while ACC1 phosphorylation was clearly detected at the 1-hour time point, it also increased with time and reached the highest level at the 24-hour time point. Therefore, we decided to use the 24-hour incubation time to achieve maximum AMPK activity.

The reviewer is right to point out that a long incubation time may increase the secondary phosphorylation events that are not directly caused by AMPK activation, which is the reason that we used the conserved AMPK substrate motif to filter out the sites that may not be directly phosphorylated by AMPK. Even after the data filtration, we still need to perform a series of experiments on these

candidate sites, like we did for ARMC10, to further validate that these candidates are truly direct AMPK substrates.

7. Fig. 1c: what concentration of A-769662 was used?

We used 100 μ M A-769662 for the experiment presented in Figure 1c. This information is now added to the legend of Figure 1c in the revised manuscript.

8. Fig. 1d: what incubation time and concentration of A-769662 – although this information is in the Methods section, it would be very helpful if it was also in the legend.

Thank you for the suggestion. We moved this panel to Supplementary Figure 1c as suggested by another reviewer. We also modified the legend of Supplementary Figure 1c in the revised manuscript as “Cells in light or heavy medium were treated with or without 100 μ M A769662 for 24 hours.”

9. Supp. Fig. 1: the Figure (for which little background information is supplied) seems to imply that HEK-293 and U2OS cells contain only two alleles, but according to the CanSAR website these cell lines are hypotriploid and hypertriploid respectively, so that there may be more than two alleles.

To verify the KO cells, we first performed Western blots to make sure the protein expression was not detectable. We then extracted the genomic DNA, amplified the targeted region by PCR, and cloned the PCR products into T vector. Twelve clones were selected for sequencing based on white-blue plaque selection. We identified only two mutated or deleted forms from those twelve sequence results. It is possible that these cells may be hypotriploid or hypertriploid, but only two mutations or deletions were identified, which means some of the mutated alleles may share the same changed sequence. The most important thing is that no wild-type sequence was recovered. Furthermore, for the AMPK-DKO cells, we verified the phosphorylation of some known AMPK substrates in both AMPK WT and DKO cells as shown in Figure 1b to ensure that AMPK activity was abolished in these DKO cells.

10. Fig. 2c: they mention in the Discussion that Ingenuity software was used for this analysis, but that information should be in the legend and Methods section. Also, it would be very useful to have a table where the proteins included are listed, along with their functional annotations.

Thank you for the suggestions. We added more description to Results, Methods, and Figure legends in our revised manuscript. We generated a new table, Supplementary Table 3, which includes the functional annotations for each gene.

11. Fig. 2d: why was LIPE not included in this list? Note also that the site on CTNNB1 does not really fit the AMPK consensus motif. Also, please check the residue numberings in this list (and in Supp. Fig. 2) carefully, as they do not always appear to correspond to the numbering in the current UNIPROT entry for that protein (e.g. NUMA1, REEP1). Note that MaxQuant sometimes assigns residue numbering using older database entries. Also, which paper previously identified S74 on MAST2? Although MAST2 is identified in Ducommun et al, they did not identify the actual site.

Thank you for the questions. We didn't include the LIPE site because this site was reported in the cow. The phosphorylation site S554 is highly conserved in humans as the site S855. We now included this site in Figure 2d of the revised manuscript as a low throughput-identified AMPK phosphorylation site.

We used two rules to filter the candidate AMPK substrates: 1) if a peptide contained both major conserved sites, the P-3 site as R/K/H and the P+4 site as L/V/I, it was considered a candidate AMPK substrate; 2) if a peptide contained only one of the two major conserved sites, we then determined whether it had any of the other three conserved sites, including the P-5 site as L/M/I, the P-4 site as R/K, and the P+3 site as N/D/E. If yes, we also considered it as a candidate AMPK substrate. CTNNB1 was selected because it has R at the P-3 site and R at the P-2 site. Therefore, we included this site even though the P+4 site does not match to L/V/I.

Thank you for pointing out the issue regarding residue numbering. The reviewer is right that many of the residue numbers are not correct, because MaxQuant sometimes assigns residue numbering using older database entries. We made these corrections in Figure 2d, Supplementary Figure 2a, and all the Supplementary Tables in the revised manuscript.

We carefully went over the phosphosites in Figure 2d and generated a new list for known AMPK substrates. We removed the S74 on MAST2 from Figure 2d in the revised manuscript.

12. Supp. Fig. 2: what program was used to search the hits for the AMPK motif, and what flexibility in the motif was allowed? Some of the sites do not appear to be good fits for the AMPK consensus motif.

Thank you for the question. The consensus motif was derived from Hardie et al. (Trends Cell Biol., 2016) and was also shown by Gwinn et al. (Mol. Cell., 2008). We aligned our candidate AMPK substrates with the consensus motif using multiple sequence alignment. We manually examined the sequence alignment using the following rules: 1) if a peptide contained both major conserved sites, the P-3 site as R/K/H and the P+4 site as L/V/I, it was considered a candidate AMPK substrate; 2) if a peptide contained only one of the two major conserved sites, we determined whether it had any of the other three conserved sites, including the P-5 site as L/M/I, the P-4 site as R/K, and the P+3 site as N/D/E. If yes, we also considered it as a candidate AMPK substrate.

We have now revised the motif searching description in Results section to make it clear.

13. Lines 132-133: to avoid confusion, please give individual references to each target protein rather than grouping the references.

Thank you for the suggestion. We have now put individual references to each target protein in the revised manuscript.

14. Lines 143-146: is the site they identify on KLC4 equivalent to the S539 or the S575 site on KLC2?

The site we identified on KLC4 is very similar to the S575 site on KLC2. The S590 phosphosite sequence of the KLC4 site is MKRAApSLNYLN, while the S575 phosphosite sequence of the KLC2 site is MKRASpSLNFLN. This information has been added to the revised manuscript.

15. Fig. 3a: please explain in the legend what MFIN (S45)_1 and MFIN (S45)_2 refer to.

We apologize for this confusion. As stated, MFIN (also called ARMC10) has two isoforms. MFIN(S45)_1 is the phosphopeptide from the short isoform of MFIN, which is 308 residues. MFIN(S45)_2 is the phosphopeptide from the long isoform of MFIN with 343 residues. We modified the figure legend of Figure 3a to include this information.

16. Fig. 3b: there does seem to be less signal with the P-Ser antibody for MFIN, KLC4 and REEP2, and with the p-AMPK substrate motif antibody for MFF, in the AMPK KO cells. However, the FLAG blots in the lower panel are over-exposed, and it is possible that in some cases there is also less protein in the AMPK KO cells.

In this Western blot, we used P-Ser antibody. This antibody can detect all phosphorylated serines in the sample, which may show high background if the protein has multiple serine sites phosphorylated by other kinase(s). The reviewer is correct that the input may also have less protein in some samples from AMPK KO cells. We did not perform any detailed quantification on this Western blot, since these candidates need to be further validated anyway using AMPK-specific assays, such as the *in vitro* AMPK kinase assay and *in vivo* AMPK dependent assay using site-specific phospho-antibodies as we conducted for ARMC10.

17. Fig. 3c (also Suppl. Fig. 3a) : what does the color coding mean?

The color coding indicates the conserved residues near the AMPK phosphorylation site. We made the color consistent in Figure 3c and Supplementary Figures 2a and 3a in the revised manuscript.

18. Fig. 3d: to enable the reader to understand what had been done, much more detail about the tags utilized, and the immunoprecipitation protocol used, needs to be provided in the Figure legend and/or Materials & Methods.

Thank you for the suggestion. We added more details about the experiment in the figure legends of Figure 3d.

19. Fig. 3e: the effect of A769662 on phosphorylation of S45 is quite modest – it would be helpful to also have a pACC blot here as a positive control, to see how this compared.

Thank you for the suggestion. As this figure does not add a lot additional information, we removed it from the revised manuscript.

20. Fig. 3f: the alignment of the ³²P autoradiogram with the blots seems to be faulty – it appears from this Figure that MFIN is phosphorylated better in the absence of AMPK! There is also a (less serious) problem with Fig. 3g, where it is probably only the “+/-” labels that are poorly aligned. Also, in the legends it states that “both isoforms of MFIN were phosphorylated by the AMPK activator A769662”, but A769662 is not an agonist, not a protein kinase!

Thank you for the comments. Figure 3f has been relabeled as Figure 3e, and Figure 3g has been moved to Supplementary Figure 4a in the revised manuscript. We re-aligned the ³²P autoradiogram with the blots in Figure 3e. We also corrected the alignment of the “+/-” labels in Supplementary Figure 4a. For the figure legends, we revised this to “Both isoforms of ARMC10 (MFIN) were phosphorylated by the AMPK complex at S45 site”.

21. Supp. Fig. 3b: this Figure is not referred to in the text, and it is not clear what it adds.

Supplementary Figure 3b depicts the cell extracts we used to validate the phospho-AMPK substrate motif [LXRXX(pS/pT)] antibody. It is not an informative panel, since the same information is available in Figure 1b or 1d. We deleted this panel in the revised manuscript.

22. Supp. Fig. 3c: it seems strange that the phospho-AMPK substrate motif antibody seems to detect many proteins in the lysate, but detects none of them after immunoprecipitation. Incidentally, it is presumed that some of the bands in these blots represent the heavy and light chains of the immunoglobulin used – it would have been helpful if these were labelled. Finally, is this Figure worth including at all?

Thank you for the suggestion. Supplementary Figure 3c has been re-labeled as Supplementary Figure 3b. The figure was designed to prove that the phospho-AMPK substrate motif antibody is generated for substrates with sequences like [LXRXX(pS/pT)]. This phospho-specific antibody recognizes only a subset of AMPK substrates (for example, MFF1 as we showed in Figure 3b). The phospho-AMPK substrate motif antibody can detect many proteins in the lysate, but no band after pulldown. This may be due to several factors: the antibody may not be effective in pulldown, the antibody used in pulldown may not be enough because many proteins can bind to that antibody, or each protein that has been pulled down by this antibody is present at a level too low to be detected or needs a much longer exposure time to show up on the film.

We labeled the immunoglobulin heavy and light chains. This figure may be informative to other researchers who want to try this AMPK substrate motif antibody. We can delete it if the reviewers find it unnecessary.

23. Lines 193-195 and Figs. 4b: it appears that only the smaller isoform of MFIN is phosphorylated in intact cells – is that correct?

According to our phosphoproteomic data, which was generated using HEK293A cells, we identified two different phosphopeptides with the same S45 site that are significantly different between AMPK WT and AMPK DKO cells. One is from the shorter isoform, and the other one is from the longer isoform. Therefore we believe both isoforms are phosphorylated in the cell. The reason we cannot see the phosphorylation of the longer isoform may be due to the fact that the longer isoform is much less abundant than the shorter isoform, as shown in Figure 3f and Supplementary Figure 4d. Another reason is that the phospho-antibody we used was generated using the peptide GIRSSKpSAED, which is derived from the shorter isoform. The longer isoform has a slightly different sequence (GIRSSKpSAGA). It is possible that this phospho-antibody may preferentially recognize the phosphorylated shorter isoform in the cell. This speculation has been added to the revised manuscript.

24. Lines 212-215 and Fig. 4c/d: they state that “we observed drastic changes in mitochondrial morphology only in cells with WT MFIN overexpression (Fig. 4c)”. While there does seem to be a difference in mitochondrial morphology between over-expression of WT MFIN and the S45A mutant, they do not analyse control (untransfected) cells in Figs. 4c or 4d, so you cannot really conclude that

there is a drastic change on over-expression of the WT. Also, in Fig. 4e there does not seem to be much change in mitochondrial morphology on over-expression of the wild type.

Thank you for pointing this out. The experiment shown in Figure 4c/d was performed in HEK293T cells. We compared and quantified mitochondrial morphology changes only between ARMC10 (MFIN) WT and ARMC10 (MFIN) S45A overexpression cells, not with untransfected control cells. Therefore, we revised our statement in the revised manuscript: “We observed drastic changes in mitochondrial morphology in cells with WT ARMC10 (MFIN) overexpression when compared with those with overexpression of ARMC10 (MFIN) S45A mutant”. When we observed this interesting phenomenon, we changed to HEK293A and U2OS cells for more detailed analysis, since HEK293T is not easy to handle for immunofluorescence and confocal assays.

25. Fig. 4d/f: here they have divided mitochondria up into four morphological categories, but there are no details of how this was done (e.g. by manual inspection, or using software?). Also, when they do statistical analysis to compare columns, which categories of mitochondrial morphology (e.g. long, short, fused or fragmented) are they comparing? More explanation is needed.

The details of these experiments have been added to the Methods section. The quantification was done with three replicates and 150 cells counted for each replicate. Cells were counted via manual inspection by a person who was blinded to genotype and treatment. We used Long versus Short morphology for comparison in our statistical analysis.

26. Figs. 4a/b/g: these panels really seem to belong more in Fig. 3 (Validation of AMPK substrates) rather than Fig. 4 (Function assay of MFIN and MFIN phosphorylation at S45).

Thank you for the suggestion. We moved these panels to Figure 3 or Supplementary Figure 4 and revised the figure legends and Results section accordingly.

27. Fig. 4a: the difference in MFIN phosphorylation between lanes 1 and 2 is modest, and the FLAG and tubulin blots are over-exposed so it is difficult to say that the amount of protein loaded is the same.

We replaced the FLAG blot with a blot exposed for a shorter time (Figure 4a was moved to Supplementary Figure 4c as suggested). It shows that the loading amount in lane 2 was not higher than that in lane 1, maybe even a little bit less. But the phosphorylation of ARMC10 (MFIN) in lane 2 has stronger signal, supporting the idea that activation of AMPK by A769662 led to increased phosphorylation of ARMC10 (MFIN).

28. Fig. 4e/f and lines 235-238: they state that cells expressing the S45E mutant had increased mitochondrial fission and fragmentation, but according to Fig. 3f there does not appear to be any significant increase in the proportion of fragmented mitochondria.

Thank you for the question. In this sentence, what we wanted to say is that there was an increase in mitochondrial fission. For mitochondrial fission, the mitochondrial shape changes from a long to a short or fragmented form. We attempted to use fragmentation to describe these mitochondrial shape changes, which include both short and fragmented forms. To avoid further confusion, we delete the word “fragmentation” from our revised manuscript.

29. Fig. 4g: because there is quite a high basal MF1N phosphorylation in the WT cells, it would be good to see a pACC blot as a positive control.

Thank you. We added P-ACC blot as a control in the revised Figure 3f.

30. Fig. 4c: if they were to provide unmerged images, this would allow the reader to more easily compare mitochondrial morphology in transfected and untransfected cells.

Thank you for the suggestion. We included unmerged images in the original Figure 4c, which is now Figure 4a in the revised manuscript.

31. Supp. Fig. 4c: there is no quantification of these results and no indication of how many times the experiment has been performed.

Thank you. We have now added the quantification and experimental details in the revised manuscript. This figure is now shown as Supplementary Figures 5c and 5d in the revised manuscript.

32. Supp. Fig. 4d: there is no explanation provided of how this Table was arrived at, or what the numbers mean.

We moved Supplementary Figure 4d to Figure 6a in the revised manuscript. In this figure, we show how we used the BioID2 method to discover ARMC10-binding proteins. We used two controls, a BioID2 tag only and a BioID2 tag fused with another protein—for comparison. The numbers are PSMs identified in BioID2_ARMC10 or BioID2_control. The p values were generated by statistical analysis of identified PSMs of the BioID2-ARMC10 binding proteins comparing to the those identified from control samples. We added more details in the figure legends and Methods section of the revised manuscript.

33. Fig. 5a/b: to my eyes, mitochondrial morphology looks different in Fig. 5a in the WT and KO after DMSO treatment, whereas in the quantification in Fig. 5b they look almost identical.

We apologize. The pictures we presented were randomly selected to show the mitochondrial morphology in these cells. We did not pay close attention to these images, since we can find mitochondria with different morphology in both WT and KO cells. This is the reason that quantification is important to draw any conclusion on changes of mitochondrial morphology. We have now replaced these images with ones better reflecting our quantification results.

34. Lines 252-256: this sentence equates glucose starvation with energy stress, whereas it has been shown that, depending on the cell type, glucose starvation is not always associated with changes in adenine nucleotide levels, although AMPK is still activated [Zhang et al (2017) Nature 548:112]. I suggest replacing “energy stress” simply with “glucose starvation”.

Thank you for the suggestion. We replaced “energy stress” with “glucose starvation” in the revised manuscript.

35. Fig. 6a: this is an example of a Figure where more documentation of the experiment is required.

Thank you for the comment. We added more details in the figure and figure legend.

36. Figs. 6a/b: why does MFF appear as one band when pulled down with SFB but at least three bands when pulled down with anti-myc?

Thank you for the question. These figures, which show pulldown with anti-Myc for the gene MFF, were relabeled as Figure 6c and Supplementary Figure 5e in the revised manuscript. The overexpression of SFB-MFF was pulled down by streptavidin beads and detected by anti-Flag antibody. The only band in pulldown with SFB was SFB-tagged MFF. The overexpression of Myc-MFF was pulled down by anti-Myc antibody and detected by anti-Myc antibody. The lysates of cells overexpression Myc-MFF show binding to two protein bands, and the pulldown figure shows three. The band around 37 KDa is Myc-MFF. The band in the pulldown figure run at 50 KDa is the antibody heavy chain band. However, the third band between 50-75 KDa, which shows in both cell lysis and pulldown figures, is not clear. We regenerated the Myc-MFF construct and redid the pulldown experiments. This band was gone in the repeat experiment, as shown in Figure 6c and Supplementary Figure 5e. The band in the original pulldown experiment may have been due to contamination, but this problem was resolved in the repeat experiment, and the revised manuscript reflects this.

37. Fig. 6e: are there arrows missing from this model? More explanation is required.

Thank you for the comment. We added the arrow back in the revised Figure 6d (labeled as Figure 6e in the original manuscript). We also added more description about the model in the revised figure legend.

38. Fig. 6d: why does myc-tagged MFIN and MFF appear to run at 100 kDa, whereas in 6b and 6c it runs at around 40 kDa?

Thank you for that question. We added more details to these figure legends to make the descriptions clear and more accurate. In Figure 6c (previously labeled Figure 6d), the Myc-tagged ARMC10 (MFIN) and MFF appear to run at around 40 KDa, and the 100-KDa protein is SFB-tagged Drp1. In Supplementary Figure 5e (previously labeled Figure 6b), ARMC10 (MFIN) is tagged with SFB, and MFF is tagged with Myc. And in Supplementary Figure 5f, ARMC10 (MFIN) is tagged with Myc, while MFF is tagged with SFB. SFB (S protein tag-Flag tag-Streptavidin binding tag) triple-tag is much bigger than Myc-tag, and the molecular weight of these tagged proteins are correct.

39. The Discussion section is rather repetitive of the Results section and could be condensed – it should discuss the results rather than just restating them.

Thank you for the suggestion. We revised the Discussion section in the manuscript.

40. Lines 367-369: the authors claim that over-expression of MFIN in U2OS cells led to mitochondrial fragmentation, but I could not find any Figures to show this.

Thank you for the comment. The overexpression of ARMC10/MFIN was done in HEK293T and HEK293A cells. In the revised manuscript, we included the overexpression of ARMC10 in U2OS cells as shown in Figure 5a.

42. Lines 377-379: it would be more appropriate to say that AMPK is regulated by AMP and ATP, rather than ATP alone.

Thank you for the comment. We revised the manuscript as suggested.

Reviewer #2 (Remarks to the Author):

Ncomms-18-15823 Global phosphoproteomic analysis reveals MFIN as an AMPK substrate that regulates mitochondrial dynamics

AMPK is a master regulator of cellular energy homeostasis. It is a heterotrimeric complex with one catalytic subunit (alpha) and two regulatory subunits (beta and gamma). The goal of this work is to identify AMPK substrates by comparing global quantitative proteomic phospho-peptides in wild type and AMPK knock out cells following activation of AMPK by a small molecule A769662, which prevents dephosphorylation of Thr172 on the alpha subunit.

Of 32 AMPK phosphopeptides identified in the analysis, twenty were novel. One was chosen for a more extensive analysis: ARMC10 (MFIN). This protein is localized to the mitochondrial outer membrane and authors argue that it plays a AMPK-regulated role in mitochondrial division. The validation of ARMC20/MFIN as an AMPK substrate is strong; however, the data quantifying mitochondrial morphology and the role of this pathway in cell death does not support the conclusions at this time.

Thank you for the nice summary and helpful comments. Please see below for our responses to your suggestions.

Specific comments:

In Figure 3b, five potential protein targets of AMPK are overexpressed in wild type or AMPK DKO HEK cells treated with A769662, immunoprecipitated and subject to western blot analysis with a phospho-Serine or P-AMPK substrate antibody.

The P-AMPK antibody only recognized Mff, a previously reported substrate of AMPK. The P-Ser antibody has signal for MFIN, KLC4 and REEP2. Therefore, REEP1 was not detected by either antibody.

In all cases, there is significant signal in the double knock out cells, which should lack AMPK activity.

Authors should comment on the source of the phosphorylated protein in these cells. Alternatively, an alanine mutant could be generated for each predicted site and expressed as in Figure 3e. As presented, the data are not a compelling argument that the new targets are bona fide, and could be removed or moved to supplement.

Thank you for the comment. As mentioned earlier in this response, the pulldown-Western blotting assay was conducted using P-Ser antibody. This antibody can detect all phosphorylated serines in the sample, which means that it would show high background signal if the protein has multiple serine sites that are not only phosphorylated by AMPK but also by other kinases in the cell. Figure 3b is just the first step of phosphorylation validation, and these candidates need to be further confirmed in future studies using AMPK-specific assays, such as the *in vitro* AMPK kinase assay and the *in vivo* AMPK dependent assay using site-specific phospho-antibodies, as we conducted for ARMC10. Yes, an alanine mutant form would help in validation of the phosphorylation sites. We generated ARMC10 S45A mutation for validation; the results are shown in Figure 3e and Supplementary Figures 4a, 4b, and 4c.

In Figure 3d, interaction of overexpressed MFIN is immunoprecipitated from cells and to test for a physical interaction with AMPK-alpha, CAMKK-beta and LKB1. Authors should discuss how CAMKK-beta and LKB1 were chosen as negative controls. For example, do these often share substrates with AMPK?

Thank you for the question. Our working hypothesis is that ARMC10 is an AMPK phosphorylation substrate. We suspect that ARMC10 binds to AMPK. We selected CAMKK-beta and LKB1 for two reasons: 1) both of these are also protein kinases, and 2) they are upstream kinases of AMPK, having the capacity to phosphorylate AMPK and regulate its function. They normally do not share substrates with AMPK. We include this information in the revised figure legend.

Phosphoblock mutants and in vitro kinase activity assays suggest that MFIN/ARMC10 is a bona fide AMPK substrate. In Figure 3f, the individual lanes in the three panels of the gel are not aligned, making it difficult to interpret the data and raising the possibility that these are not from the same experiment or gel.

We acknowledge the error. We have re-aligned the ³²P autoradiogram with the blots in Figure 3e (original Figure 3f). They are from the same experiment. The upper film is the ³²P autoradiogram, and the other two, lower films are Western blots for ARMC10 and AMPK α , which are the loading controls for the kinase reactions.

Both the short and long isoform of the protein is included in Figures 3f, 3g and 3h, but this is not described clearly in the text.

Thank you for the comment. We added “The two phosphopeptides we identified for ARMC10 S45 site are the same phosphorylation site in two different ARMC10 isoforms. The shorter isoform is 308 residues, and the longer isoform is 343 residues. We generated the WT and ARMC10 S45A mutants for both isoforms” to the revised manuscript to make it clearer.

Overexpression of N-terminally tagged MFIN/ARMC10 in HEK293T cells appears to cause mitochondrial fragmentation, while the S45A mutant does not.

Vector controls are lacking from this experiment and make it impossible to determine the degree to which the mitochondrial morphology is altered in cells expressing MFIN/ARMC10.

We initially quantified mitochondrial morphology changes between MFIN WT to MFIN S45A in HEK293T cells and did not include vector control in this study. When we discovered that ARMC10 may function in mitochondrial fusion and fission, we switched to HEK293A and U2OS cells for more detailed analysis since HEK293T is not easy to handle for immunofluorescence staining and confocal assays. We included vector controls in experiments using HEK293A and U2OS cells (please see Figures 4c/d and 5a/b).

The metric for scoring mitochondrial morphology must be included in the material and methods. For example, how were long and fused mitochondria distinguished? How many cells were counted? How many times was the experiment repeated? Were the samples blinded?

Thank you for the questions. We added details about these experiments in the revised Methods section, including the definition of mitochondrial morphology (i.e., fused, long, short, and fragmented), the number of times the experiments were repeated, how we counted these cells, and the results.

To determine if the phosphorylation state at Ser-45 alters the activity of MFIN/ARMC10, the phosphomimetic mutant S45E is expressed in wild type and AMPK DKO cells. Activation of AMPK activity in wild type cells induces mitochondrial fragmentation, but not in the DKO cells. Regardless of AMPK activation, cells expressed MFIN/ARMC10 S45E are scored as having a higher proportion of cells with fragmented mitochondria.

The text is not consistent with the metric used on the graph, where the increased population is referred to as “short”.

This analysis also lacks a vector control for reference.

Thank you for the comments. The term “mitochondrial fragmentation” we used to mean mitochondrial fission, including both short and fragmented mitochondria. To avoid any misunderstanding, we changed “mitochondrial fragmentation” to “mitochondrial fission” in the revised manuscript.

The immunostaining pattern is odd in these experiments. Tom20, a mitochondrial outer membrane protein, is labeled with green and MFIN/ARMC10 is labeled with red. In contrast to the wild type and S45A variants, the S45E variant does not appear to completely co-localized with Tom20. Specifically, the perinuclear mitochondria are labeled only with Tom20 and the peripheral mitochondrial with MFIN/ARMC10 S45E. It looks like the Tom20 signal is mitochondrial aggregates but the MFIN/ARMC10 are not. This raises the possibility that the tag or the mutation is altering the function of MFIN/ARMC10 and may not represent an accurate assessment of protein function.

Yes, from the images we originally presented, some of the region looks green. We checked the unmerged raw data carefully. The ARMC10 and TOM20 staining matched very well, but the distribution of staining intensity varied and therefore when we merged these images some region turned out to be closer to green, while others were closer to red. We have replaced these images with others that more clearly demonstrate our point and also presented unmerged images in Figure 4a. Please be aware that ARMC10 S45E mutant is a constitutive active mutant, which promotes mitochondria fission as well as mitophagy. When levels of ARMC10 S45E expression were high, we observed reduced TOM20 levels in the cell.

CRIPSR-Cas9 knock out of MFIN/ARMC10 are generated in U2OS cells. Compared to controls, the knock out cells have extended/more connected mitochondria. In contrast to the wild type control, the mitochondria in the knock out cells do not fragment with activation of AMPK following A769662 treatment. These data argue that MFIN/ARMC10 plays a role in determining mitochondrial structure. The N-terminally tagged variant utilized in previous experiments should be expressed in the knock out cells to assess function, which could be altered by the tag.

To further substantiate that MFIN/ARMC10 activity is regulated, the S45A and S45E mutants should be expressed in these cells and mitochondrial morphology scored with and without A769662 treatment.

Thank you for the comments. As shown in Figure 5a of the revised manuscript, we added experiments with expression of WT, S45A, or S45E ARMC10 in the ARMC10 KO cells with or without A769662 treatment. The result prove that ARMC10 plays a role in determining mitochondria morphology changes.

Mitochondrial dynamics influence susceptibility to cell death. Cell survival was assessed following energy stress (glucose starvation) to determine if MFIN/ARMC10 plays a role in this pathway. The Figure Legend suggests that all cell lines are HEK293T; was MFIN/ARMC10 also knocked out in these cells? Assuming

that the MFIN/ARMC10 knock out cells are U2Os, a wild type U2OS cell line is required in this analysis, to demonstrate that this cell type has a comparable response to wild type HEK293T cells.

Thank you for the question. The experiment depicted in Figure 5c was done in HEK293A cells. We used this cell line to study cell survival, because we have both AMPK DKO and ARMC10/MFIN KO in this cell line, which is shown in Supplementary Figures 4e and 4f. Now, we got the AMPK DKO in U2OS cell line, a gift from Reuben Shaw group (Science, 2016). When we did the cell survival experiments in U2OS cells, we got similar results, as shown in Figure 5d.

Minor Comments:

Figure 1A lacks labels for each lane.

We corrected this in the revised figure.

Figure 1C legend should state that these are wild type cells.

We changed it to “Time course of treatment with the AMPK activator A769662 (100 μ M) was conducted in WT HEK293A cells. Western blot was conducted using antibodies as indicated” in the revised figure legend.

Authors switch between ARMC10 and MFIN throughout the manuscript. MFIN is very close to MFN, which is a mitochondrial fusion factor, and could be a confusing acronym to adopt.

Thank you for the comment. As suggested, we decided to use ARMC10 consistently in our revised manuscript.

Lines 259 – 273 discuss a connection with mitophagy, which has been previously correlated with mitochondrial fragmentation. Without additional experiments, this data seems preliminary and difficult to draw a conclusion from.

Thank you for this comment. We agree this reviewer that the mitophagy data are preliminary and limited. We included these data in a Supplementary figure since the data are reproducible and it is possible that severe or prolonged mitochondrial fission may lead to mitophagy. We concur that more experiments are needed to determine the mechanisms underlying the role of ARMC10 (MFIN) in mitophagy. We decided to include this as a Supplementary figure since it may help readers who are interested in studying the mechanism of AMPK-ARMC10 (MFIN) in mitophagy, but of course we will be happy to remove it at your or editors' request.

Reviewer #3 (Remarks to the Author):

In the manuscript titled ‘Global Phosphoproteomic Analysis Reveals MFIN as an AMPK Substrate that Regulates Mitochondrial Dynamics’, the authors discuss the use of a double knockout cell line to identify phosphosites directly or indirectly regulated by AMPK, followed by the validation of identified phosphosites and investigation of the MFIN interactome. Although the work presented is of interest it

still requires a major rewrite and additional analysis, mainly related to the clarification of the experimental setup and proteomics data analysis. Please find specific comments below.

Thank you for the nice summary and helpful comments. Please see below for our responses.

Main comments

1. The authors could benefit from a more detailed description of the reasoning behind their experiments. Currently, it is not clear why certain experiments were performed or why certain protein candidates were chosen for further experiments. For example, the importance of Fig 1C is not discussed in the main text nor figure legend, whereas it is probably the basis for the decision to stimulate cells with A769662 for 24 hours (mentioned much later in the text).

Thank you for the comment. We added more details in the revised manuscript, including the discussion of Figure 1C. "Phosphorylation of Thr-172 in the activation loop of AMPK α is indicative of AMPK kinase activity. As shown in Fig. 1c, T172 phosphorylation increased with time and was the greatest at the 24-hour treatment time point. Similarly, while ACC1 phosphorylation was clearly detected at the 1-hour time point, it also increased with time and reached the highest level at the 24-hour time point. To maximize our ability to detect AMPK-dependent phosphorylation sites in comparison, we therefore treated both AMPK WT and AMPK α 1/ α 2-DKO cells with A769662 100 μ M for 24 hours after stable isotope labeling by amino acids in cell culture."

2. The authors do not clearly specify their bioinformatics workflow. At present it is unclear whether they calculate ratios between phosphosites or identified phosphopeptides, how many unique and high-confident phosphosites were identified (as they only indicate the number of phosphopeptides) and the cutoffs used for filtering the data are not supplied. The authors should specify cutoffs used for localization probabilities, identification in biological replicates, and so on. The use of a median in the low number of replicates is questionable.

Thank you for these comments. In the revised manuscript, we added more details in the Methods section about the quantification process and phosphosite assignment. We clearly indicate that ratios between identified phosphopeptides were calculated. The phosphosites' localization probabilities are included in the revised Supplementary Table 1 and Table 2. We also indicate that 72.16% of the phosphopeptides had a localization probability higher than 0.7 for the phosphosite assignment. The quantification cutoff values were fold-change greater than 1.5 and p value less than 0.01 (by t-test). The p value was calculated among the four biological repeats, and it may indicate how it is repeatable. All of the quantification and localization probability information can be found in the revised Supplementary Table 1 and Table 2. Description of these approaches was added to the Results section of the revised manuscript.

3. When investigating MFIN (Fig 3F, 4B, 4G) it seems that the MFIN-343-S45A mutant is still phosphorylated by AMPK in vitro (Fig. 3F). When assessing phosphorylation in cell lines, it appears only one of the two isoforms of MFIN is actually phosphorylated (Fig 4B and 4G), as indicated by a double MFIN band versus a single Phos-MFIN band. Yet both isoforms were reported as phosphorylated based on the phosphoproteomics study. This discrepancy is confusing and should be discussed.

Yes, we detected a very weak band for ARMC10 (MFIN)-343-45A mutant, shown in Figure 3e (original Figure 3F), indicating that AMPK may be able to phosphorylate additional site(s) on ARMC10, at least *in vitro*. However, we did not observe any additional AMPK-dependent phosphorylation sites on ARMC10 based on our mass spectrometry results, suggesting that any additional phosphorylation may not exist or be very weak *in vivo*. Therefore, we did not further investigate this weak phosphorylation site.

From our phosphoproteomics data, we identified two different phosphopeptides with the same S45 site that are significantly different between AMPK WT and AMPK DKO cells. One is from the shorter isoform, and the other one is from the longer isoform. Therefore we believe both isoforms are phosphorylated in the cell. The reason we cannot see the phosphorylation of the longer isoform may be the fact that the longer isoform is much less abundant than the shorter isoform, as shown in Figure 3f and Supplementary Figures 4d and 4e. Another reason is that the phospho-antibody we used was generated by using the peptide GIRSSKpSAED, which is derived from the shorter isoform. The longer isoform has a slightly different sequence (GIRSSKpSAGA). It is possible that this phospho-antibody may preferentially recognize the phosphorylated shorter isoform in the cell.

4. On page 14, line 277 and onwards, the authors discuss a BioID experiment where they investigate the MFIN interactome. The setup of this experiment is not clear, please specify control bait protein, experimental conditions and elaborate on data analysis leading to Supplementary Figure 4D. Also, in Supplementary Figure 4D, the meaning of numbers listed under the columns “BioID2_MFIN” and “control” is unclear.

We moved this figure to Figure 6a in the revised manuscript. In this experiment, we used the BioID2 method to discover ARMC10-binding proteins. We used two controls, a BioID2 tag only and a BioID2 tag fused with another protein—for comparison. The numbers are PSMs identified in BioID2_ARMC10 or BioID2_control. The *p* values were generated by the statistical analysis comparing the identified PSMs of the BioID2-ARMC10 binding proteins to those identified from control samples. We added more details in the figure legend and Methods section of the revised manuscript.

General comments

1. The discussion is currently resembling a summary of the data, we feel the authors could make better use of the discussion by discussing the major drawback of their technique, namely the risk of comparing a dysregulated/stressed cell line with a WT cell line and the inability to distinguish whether differences in phosphosites are due to the absence of AMPK or a general stress response in the cell.

Thank you for this suggestion. We revised the Discussion section of the manuscript. Four technical limitations of this strategy are now fully discussed.

2. On page 4 line 70, the authors state that “51 phosphosites were found to have lower levels in AMPK WT cells than those in AMPK DKO cells, suggesting that these are likely phosphorylation events negatively and probably indirectly regulated by AMPK”, however, since the authors generated a permanent DKO of a major kinase complex, these sites may very well reflect some of the cell’s compensating mechanisms. This possibility should be addressed by the authors. This possibility is reflected in Supplementary figure 1A, where the knockout of AMPKa1 clearly resulted in an upregulation of AMPKa2 and vice versa.

Thank you for your comments. Yes, we agree with this reviewer that compensating mechanisms may have existed when we generated permanent DKO of AMPK α 1/ α 2. It could be a very interesting project to further study any potential compensation and its underlying mechanisms in AMPK DKO cells. In this study, we revised our statement as “Another 51 phosphosites were found to have lower levels in AMPK WT cells than those in AMPK DKO cells, suggesting that these are likely phosphorylation events negatively and probably indirectly regulated by AMPK expression.” In addition, we include discussion about potential compensating mechanisms in the Discussion section, as mentioned above.

3. On page 4, line 72, the authors use the “known conserved AMPK phosphorylation motifs” to distinguish directly and indirectly regulated AMPK substrates, however, the authors do not specify what they define as the conserved motif. Moreover, filtering on known motifs could hamper the identification of new phosphosites or substrates, especially when the known substrates do not share a strict consensus sequence (e.g. for AMPK, the phosphosite database revealed 65 known substrate sequences, without very strict amino acid conservation between them).

The consensus motif was derived from Hardie et al. (Trends Cell Biol., 2016) and also was shown by Gwinn et al. (Mol. Cell., 2008). We aligned our candidate AMPK substrates with the consensus motif using multiple sequence alignment. We manually examined the sequence alignment using the following rules: 1) if a peptide contained both major conserved sites, the P-3 site as R/K/H and the P+4 site as L/V/I, it was considered a candidate AMPK substrate; 2) if a peptide contained only one of the two major conserved sites, we determined whether it had any of the other three conserved sites, including the P-5 site as L/M/I, the P-4 site as R/K, and the P+3 site as N/D/E. If yes, we considered it a candidate AMPK substrate. We have revised the description of motif searching in the Results section to make it clearer.

We agree with this reviewer that filtering using known motifs may hamper the identification of new substrates. We think there are two steps when dealing with this type of data. First, we align the candidates with the known motif. The matched peptides are more likely to be true substrates. We can test this group of genes first. Second, for the unmatched peptides, we will need further experiments (for example, *in vitro* kinase assays) or use another dataset (for example, phosphoproteomics data on cells responding to energy stress with or without AMPK inhibition) to help us select potential candidates for further investigation. More work will be needed to validate that these unmatched genes/proteins are indeed direct AMPK substrates, but it will provide us new AMPK phosphosites or substrates.

4. On page 9, line 181: The authors indicate that the MFIN S45A mutant was barely detectable with phospho-S antibody. It is unclear whether these HEK cells still express WT MFIN. If not, the presence of some phospho-S signal on MFIN S45A implies that there is a second, less abundant phosphorylation, which could mean the interaction between MFIN and AMPK is based on a different site altogether. In addition, the authors do not state which of the isoforms of MFIN was transiently expressed (or if both were expressed simultaneously). In Figure 3D, the authors only show AMPK α 1, while AMPK α 2 would also be informative.

Thank you for the comments. We used wild-type HEK293T cells for this experiment. The endogenous ARMC10/MFIN is still expressed in these cells. However, the molecular weights for endogenous ARMC10/MFIN and the transfected ARMC10/MFIN (WT or S45A mutant) were different, since the transfected ARMC10/MFIN S45A had an SFB tag. The weak band in the ARMC10/MFIN S45A lane may have come from other serines on ARMC10/MFIN, which may be phosphorylated by AMPK or more likely

other kinases. We did not map the interaction site between ARMC10 and AMPK. It is likely that the interaction between ARMC10/MFIN and AMPK was mediated by other residues, but not the S45 phosphorylation site.

We used the short isoform for this experiment. Accordingly, we revised the figure legend as “The phosphorylation of overexpressed short isoform of ARMC10/MFIN WT or ARMC10/MFIN S45A were detected by an anti-phospho-Ser antibody before and after treatment with the AMPK activator A769662”.

Since AMPK α 1 and AMPK α 2 have redundant functions, we used only AMPK α 1 to show that the catalytic subunit of AMPK can bind to ARMC10. As the reviewer suggested, we included AMPK α 1 and AMPK α 2 in the revised figure (Figure 3D in the revised manuscript). Both AMPK α 1 and AMPK α 2 can bind to ARMC10.

5. In figure 2A, the significance of the colors is not specified. Also, MFIN is indicated with a larger diameter dot, the meaning of which is also not specified. How authors move from identified phosphopeptides to phosphosites is not clear.

In Figure 2A, we included the information about the known substrates shown in Figure 2D, using the same colors. We include the description of the colors in the revised figure legend.

The larger diameter dot of ARMC10/MFIN is to emphasize the gene we selected for further functional study. But we agree with the reviewer that this may have been confusing to readers. We therefore changed the dots back to blue color and to the same size as others in the revised figure; instead, we simply added arrows to indicate ARMC10 peptides. Again, we define the arrows in the revised figure legend.

For the phosphoproteomic data analysis, the software provided additional information about the phosphosites along with the identified phosphopeptides. We expanded the description of the data analysis and phosphosite identification in the revised manuscript.

6. Figure 2C appears to be a very elegant representation of enriched GO terms and localization, however, it is hardly mentioned in the main text and the meaning of the number of dots per GO term is not specified. The authors should consider discussing the figure in a bit more detail. Also, the legend may be simplified by referring back to the groups indicated.

Thank you for the suggestions. We added more description in the Results, Methods, and figure legends in our revised manuscript. We also included Supplementary Table 3, which contains the GO analysis for each gene. We simplified the legend in this figure using group names as you suggested.

Minor comments

1. The authors could benefit from a general spell/style check, to ensure they use consistent naming (e.g. Crispr/Cas9, the use of ARMC10 and MFIN interchangeably, the use of KO and DKO interchangeably, the use of phospho Ser and phosphor Ser antibody, all of which create confusion).

Thank you. We corrected the spellings and imposed consistency on the names in the revised manuscript. We decided to use ARMC10 as the gene name. We used AMPK α 1/ α 2-DKO for double knockout and KO for single knockout throughout the manuscript.

2. Figure 1D is an important control to assess the influence of labeling, however, as it currently does not add to the story as described by the authors, it may be better situated in the supplementary figures.

Thank you for the suggestion. Figure 1D was included for several reasons: 1) It helps to illustrate our experimental design. We conducted two repeats with “Light”-labeled AMPK α 1/ α 2-DKO cells and “Heavy”-labeled HEK293A WT cells, and another two repeats with the reverse labeling; 2) It serves as quality control for our samples. After treatment with AMPK activator A769662, we observed strong signal for AMPK phosphorylation and AMPK substrate phosphorylation in HEK293A WT cells, but no signal in AMPK α 1/ α 2-DKO control cells. This is important information for readers. As suggested, we moved this information to Supplementary Figure 1c and added more description in the revised manuscript.

3. In supplementary table 1, the meaning of “positions within proteins” (column 2) is unclear. Are these the identified phosphosites? Do they all have confident phosphosite assignment? Please clarify.

We used Maxquant software to generate this table. Yes, the “positions within proteins” is the position of the identified phosphosites in the protein sequence. We corrected the positions within proteins in Figure 2d, Supplementary Figure 2, and all the Supplementary Tables, as another reviewer suggested that many of the residue numbers are not correct because the software used an older database. These sites all have phosphosite assignment scores for the localization probability when analyzed by Maxquant software. We added three additional columns generated by Maxquant into Supplementary Table 1 and Table 2 with the column titles “Localization prob”, “Phospho (STY) Probabilities”, and “Phospho (STY) Score diffs”. Additional supporting information was included in the revised manuscript.

4. In supplementary table 2, quite a few ratios of 0 are reported (e.g. for identifier P10071), which are left empty in supplementary table 1, indicating the corresponding phosphopeptides were not detected. Listing a zero in a log₂ ratio indicates the measured values are identical, which is not the case.

Thank you for the comment. Yes, those values should not be zero. We replaced those zero values with blanks, because they represent phosphopeptides not identified or quantified, in the revised Supplementary Table 2.

5. On page 6, line 114, the authors state: “51 phosphosites with lower levels in WT cells than those in AMPK DKO cells”. At present it is not clear whether the same cutoff was used as for upregulated phosphopeptides or if a different cutoff was used.

Thank you. In the revised manuscript, we added the cutoff information for the phosphosites with lower levels: “51 phosphosites with lower levels (fold-change lower than 0.667 and p value lower than 0.01 with t test) in WT cells than those in AMPK α 1/ α 2-DKO cells”.

6. The authors classify the 109 differentially regulated phosphosites in three groups, however, only group 1 is listed in Supplementary figure 2. It might be interesting to also show groups 2 and 3 and

whether the downregulated phosphosites also comply with the AMPK motif (indicating a more complex regulatory system).

Thank you for the comment. We listed phosphosites in Supplementary Figure 2 because we are interested in identifying potential AMPK phosphorylation sites. We also analyzed the motif for group 2 as shown in Figure 2b. For group 3, we only get one conserved amino acid P at the site right after the phosphosite, which is potential phosphorylation site for cyclin/cdks and/or MAPK. These data probably agree with the functions of AMPK, which may prevent cell proliferation and/or cell cycle progress to reduce energy consumption. We have added this information to Supplementary Figure 2b and discussed its implications in the revised manuscript. The sequence information for groups 2 and 3 can also be found in Supplementary Table 2.

7. In figure 2D, the authors list the known AMPK substrates that they have identified in group 1. At present, the references to the corresponding articles are missing from this table. The rationale for distinguishing low and high throughput studies is not clear and hence confusing. Does this mean that the sites indicated as low throughput have been biologically validated while the sites classified as high throughput were not?

Thank you for the question. The references for the known AMPK substrates in Figure 2d are shown in the Results section. We agree with the reviewer that the distinction between low- and high-throughput studies was not very clear. In the revised Figure 2d, we divided these AMPK substrates into three groups: validated substrates, substrates identified with high-throughput methods but not validated, and substrates that were identified but with phosphosites different than the known sites.

8. Currently, the authors do not emphasize the 20 novel phosphosites that they report. In order to find these, the reader needs to use supplementary figure 2 and manually subtract the 12 previously reported sites listed in Figure 2D. Why not clearly show the 20 potentially new sites that the authors have identified in this study?

Thank you for the comment. As suggested, we have now included a new Supplementary Table 4 for these potential new phosphosites identified in this study.

9. On page 8, line 142-144, the authors speculate around the role of AMPK phosphorylation of KLC4. This speculation would be supported by some additional information regarding sequence conservation between the two proteins or localization of the phosphosites on highly conserved domains. However, the authors do not comment on the similarity between the proteins, other than them sharing name and family. Interestingly, the authors speculate a similar biological function for KLC4 in their U2OS/Hek293 cells, however, axon-dendrite polarization might not be expected to occur in these cell lines.

Thank you for the comment. We added some additional information in the revised manuscript as “KLC2 and KLC4 have 67.19% identity between their protein sequences and the sequence surrounding KLC4 S590 site (i.e. MKRAApSLNYLN) is conserved and similar to the sequence surrounding KLC2 S575 site (MKRASpSLNFLN).” While axon-dendrite polarization may not be expected to occur in the cell lines we used in this study, KLC2 has been shown to regulate PI3K transportation. This function of KLC2 may be shared by KLC4. We included this discussion in the revised Results section.

10. On page 8, line 145, the authors chose 5 sites in group 1 for further analysis. Please specify which sites were chosen and why.

We chose five sites from five different proteins for further analysis. MFF S172, which also showed up in our study, was reported as an AMPK phosphorylation site in 2016. We decided to use MFF as a control in our experiments. ARMC10 S45 was selected because this site is at the top of the list when ranking by p value. Furthermore, we identified two different peptides for ARMC10 S45, which differ by an alternative splicing site two residues after the phosphorylation site. REEP1 S152 and REEP2 S150 were chosen because these two proteins are from the same protein family and the phosphorylation sites are within the same region with high sequence similarity (REEP1 S152 site: RLRSFpSMQDL; REEP2 S150 site: KLRSFpSMQDL). KLC4 S590 site was chosen because this site (i.e., MKRAApSLNYLN) is conserved and shares extensive sequence similarity with a known AMPK phosphorylation site in KLC2 (KLC2 S575 site: MKRASpSLNFLN). We have added this information to the revised manuscript.

11. In figure 3a, the authors show the log₂ ratio between WT and DKO cells for 5 proteins across the four replicates. This figure may benefit from showing average values and standard deviations instead of individual values for each replicate. At present, the figure emphasizes the variation between replicates. In addition, the difference between MFIN_S45_1 and MFIN_S45_2 is not clear here, and not explained in the text until page 9 line 168.

Thank you for the suggestions. We changed Figure 3a as suggested, with average values and standard deviations. We added the meaning of ARMC10/MFIN S45_1 and ARMC10/MFIN S45_2 in the figure legend and also described it when we introduce Figure 3a in the Results section.

12. On page 8, line 155, the authors discuss the validation of five selected sites by western blot. They state that detection of MFF and REEP1 is hampered “probably because of the low affinity of this phospho specific antibody for western blotting”. If this antibody is in fact low affinity, this should be visible in all conditions. More likely, the authors are experiencing a problem of specificity and/or sensitivity.

Thank you for the comment. We changed the description in the Results section to “We did not detect REEP1 probably because of the sensitivity of this phospho-Ser antibody for Western blotting. We also failed to detect MFF phosphorylation using this antibody. Besides the sensitivity of the phospho-Ser antibody, the expression level of MFF was low when compared to other tested genes/proteins. We found that in the case for MFF, phospho-AMPK substrate motif antibody, which is more specific than the phospho-Ser antibody, was able to detect MFF phosphorylation.”

13. In figure 3B, the indications of AMPKa1/a2 presence overlap, since all conditions are +/+ or -/-, perhaps a single indicator would suffice. In addition, the anti-flag antibody seems to indicate some truncations, which appear dependent on the presence of AMPKa1/a2 and are not discussed at all. In addition, it is unclear whether the P-AMPK substrate motif antibody binds the phosphorylated motif or the non-phosphorylated motif. More generally, it is unclear for this figure how the experiment was performed. Was the pulldown performed with P-Ser and P-AMPK antibodies or with the tagged proteins themselves? Please clarify.

Thank you for the suggestions. We now use a single indicator for AMPK α 1/ α 2 WT or DKO cells. The overexpressed tagged proteins sometimes also had lower molecular size bands, which were likely due to protein degradation. The low molecular size bands in Figure 3b also show some changes in level between AMPK α 1/ α 2 WT and DKO cells, but these changes are not consistent and therefore we did not discuss them in this manuscript. The P-AMPK substrate motif antibody binds to the phosphorylated motif, which has been used previously (for example, Figure 3d in Toyama et al [Science 351(6270): 275-281, 2016]). The pulldown was performed with the use of S-protein beads, because the genes/proteins were tagged with SFB tag. More details about the pulldown assay have been added in the figure legend and in the Results and Methods sections.

14. Supplementary figure 3 shows three panels, A-C, which are only explained in the legend. For supplementary figure 3A, the relevance of the colours is unclear. In 3B, it is unclear which extracts are meant and why this figure is in the manuscript in addition to figure 1B. For 3C, the meaning of the red boxes is unclear as well as the identity of many of the bands shown. Please include molecular weight markers. Also, it seems the figure serves to strengthen the point the authors make on p. 8 line 155, however, the figure is not referred to in that section.

Thank you for the comments. In Supplementary Figure 3a, the color coding indicates the conserved residues near the AMPK phosphorylation site. We made the color consistent in Figure 3c, Supplementary Figure 2, and Supplementary Figure 3a in the revised manuscript.

The original Supplementary Figure 3b showed the cell extracts we used to validate the phospho-AMPK substrate motif antibody. Because this was not an informative panel, since the same information is available in Figure 1b or 1d, we deleted this panel from the revised Supplementary Figure 3.

In the original Supplementary Figure 3c (Supplementary Figure 3b in the revised manuscript), red boxes have been added to show the phospho-protein band we detected with each phospho-antibody. We also labeled the antibody light and heavy chain bands. Molecular weight markers were added to the figure. This information also is explained in the figure legend. We also now refer to this figure in the revised manuscript.

15. In Figure 3G, it is unclear why mock transfection in presence of AMPK complex does not show any MFIN or phospho MFIN.

Thank you for the question. The original Figure 3g was similar to Figure 3f, which shows the results of *in vitro* kinase assays. These two original figures were relabeled Figure 3e and Supplementary Figure 4a in the revised manuscript. The experiment shown in Figure 3e used ^{32}P , but that shown in Supplementary Figure 4a used phospho-ARMC10 antibody for the detection of phosphorylated product. The mock lanes did not have WT or S45A mutant of ARMC10/MFIN. We just compared reaction buffer with or without AMPK complex. We have now removed “mock” and replaced it with “blank” in the revised Figure 3e and Supplementary Figure 4a to avoid any confusion.

16. In Figure 4C and 4E, scale bars are missing. In Figure 4D, 4F and 5B, it is unclear how Fused/short/long/fragmented are defined.

Thank you. We added the scale bars in original Figures 4C and 4E (Figure 4A and 4C in the revised manuscript). A detailed description of mitochondria morphology has been added to the revised Methods section and also to the figure legend.

17. On page 12 line 239, the authors mention MFIN KO U2OS cells. Based on Supplementary Figure 1B, this is actually a double KO or a conserved region was targeted to knock both isoforms out in a single step. However, this is not specified.

Thank you for the question. The ARMC10/MFIN KO U2OS cell is a single knockout cell which was generated by targeting a common region shared by both short and long isoforms. We added this information to the figure legend in the revised manuscript.

18. In Supplementary Figure 4B, the indices 2 and 3 of the zooms have been switched. Also, scale bars are missing. It is unclear whether overexpression of MFIN induces mitochondrial aggregation or that MFIN aggregates on the mitochondria upon overexpression.

We acknowledge this error. We have re-arranged Supplementary Figure 4b to the correct configuration and added scale bars to this figure. Our working hypothesis is that overexpression of ARMC10/MFIN induces mitochondrial aggregation. This is based on the observations that only a small fraction of U2OS cells with ARMC10 overexpression treated with A769662 display this mitochondrial aggregation phenotype. Of course, future studies are needed to elucidate the mechanisms underlying ARMC10-induced mitochondrial aggregation.

Thanks to all the reviewers for their thorough review and helpful comments! These changes made the revised manuscript substantially better. Please examine the revised manuscript and we are available to answer any additional questions. Thanks!

Reviewers' comments:

Reviewer #1 (Remarks to the Author):

The authors have made a genuine and thorough effort to respond to the many points I raised on the first version of the manuscript.

Reviewer #2 (Remarks to the Author):

The images in Figure 4 are very difficult to interpret. If an audience is to enthusiastically believe the results, the images must be more clearly presented. Smaller regions of the field could be utilized; why not just chose one representative cell? Furthermore, the boxed region should be displayed as it is in Figure 5, in a separate row, so that it can be large enough to see some detail. Unmerged images are always preferable, to allow the eye to see the signal intensity of each channel.

Authors argue that the Tom20 signal may be diminished by mitophagy. While it may be true that mitochondria are being subject to mitophagy with expression of S45E, this should not selectively remove Tom20. Does it? If this is the case, another antibody could be utilized.

The experiment in Figure 4b is not correctly performed. As also pointed out by Reviewer 1, quantification of mitochondrial morphology in the absence of control cells that are not expressing exogenous protein are meaningless. Modification of wording in the text is not sufficient to address this concern.

In lines 281 – 284, authors suggest that ARMC10 could play a role in fusion and fission. Given that AMPK has already been implicated in impacting division via Mff and your overexpression data presented in Figure 4b, why include fusion here? This should be removed for clarity.

In figures 4c and 4d, authors claim that cells expressing ARMC10 S45E have more fragmented mitochondria. The increase in short + fragmented mitochondria in the AMPK double knock out cells is relatively modest. Authors should address this discrepancy in the data. If ARMC10 S45E were sufficient to induce mitochondrial division, the WT and AMPK double knock out cells should look the same. One interpretation is that the ARMC10 S45E is less penetrant in the absence of AMPK directed phosphorylation of Mff.

Reviewer #3 (Remarks to the Author):

I would like to thank the authors for taking the time to carefully look at my comments, I feel the quality of the manuscript has improved due to this effort. Many of my comments have been addressed, however, I feel the BioID workflow is still not described. I urge the authors to provide details on the experimental conditions as well as data analysis, as is customary in the proteomics field and should be requested by the journal. Please find my final comments below.

For the BioID experiments, the protein used as negative control bait should be listed and the parameters used for protein identification as well as the scoring cutoffs employed need to be specified. Also, it is confusing that three baits were described in the experiment and the data of only two was shown in Fig. 6A. Please clarify whether the same sample preparation and mass spectrometric methods were employed here as described in the MS methods section.

In addition, it is still unclear how peptides carrying more than one phosphorylation site were treated and although the source of your GO-terms is now provided, there is still no information available on their fold enrichment, the background proteome used or any significance testing. Hence, the selection of the presented GO terms is still unclear.

Point-by-point response to reviewers' comments:

Reviewer #1 (Remarks to the Author):

The authors have made a genuine and thorough effort to respond to the many points I raised on the first version of the manuscript.

Thank you for the nice comment!

Reviewer #2 (Remarks to the Author):

The images in Figure 4 are very difficult to interpret. If an audience is to enthusiastically believe the results, the images must be more clearly presented. Smaller regions of the field could be utilized; why not just chose one representative cell? Furthermore, the boxed region should be displayed as it is in Figure 5, in a separate row, so that it can be large enough to see some detail. Unmerged images are always preferable, to allow the eye to see the signal intensity of each channel.

Thank you for the suggestion. As suggested, we selected one representative cell, and enlarged small representative region of that cell. We presented this selected region as we did in Figure 5, in a separate row in the revised manuscript. It shows the details more clearly.

We also agree with this reviewer that it would nice to include unmerged images that have the signal from each channel. We included unmerged images in Figure 4a, but not in Figs. 4c and 5a, because there are too many images in these figures. Therefore, we presented all the unmerged images for Figs. 4c and 5a in Supplementary Figs. 8a and 8b.

Authors argue that the Tom20 signal may be diminished by mitophagy. While it may be true that mitochondria are being subject to mitophagy with expression of S45E, this should not selectively remove Tom20. Does it? If this is the case, another antibody could be utilized.

Since our working hypothesis is that expression of S45E would lead to mitophagy, we do not think that overexpression of ARMC10 S45E mutant would selectively remove Tom20. To further test our working hypothesis, we used another mitochondrial marker Cytochrome c oxidase (or Complex IV), which is the last enzyme in the respiratory electron transport chain located in the inner mitochondrial membrane. As shown in Supplementary Fig. 5c in the revised manuscript, we found that overexpression of ARMC10 S45E mutant also reduced the level of Cytochrome coxidase, suggesting that this mutant may promote mitophagy in cells.

The experiment in Figure 4b is not correctly performed. As also pointed out by Reviewer 1, quantification of mitochondrial morphology in the absence of control cells that are not expressing exogenous protein are meaningless. Modification of wording in the text is not sufficient to address this concern.

As suggested, we repeated this experiment by adding control cells, in both IF images (Figure 4a) and mitochondria morphology quantification (Figure 4b) as shown in the revised manuscript.

In lines 281 – 284, authors suggest that ARMC10 could play a role in fusion and fission. Given that AMPK

has already been implicated in impacting division via Mff and your overexpression data presented in Figure 4b, why include fusion here? This should be removed for clarity.

Thank you for the suggestion. We used the fusion and fission to describe the mitochondria cycle. Yes, it can be confusing in this context and therefore we removed "fusion" in our revised manuscript.

In figures 4c and 4d, authors claim that cells expressing ARMC10 S45E have more fragmented mitochondria. The increase in short + fragmented mitochondria in the AMPK double knock out cells is relatively modest. Authors should address this discrepancy in the data. If ARMC10 S45E were sufficient to induce mitochondrial division, the WT and AMPK double knock out cells should look the same. One interpretation is that the ARMC10 S45E is less penetrant in the absence of AMPK directed phosphorylation of Mff.

Thank you for the comment. Yes, in this rescue experiment, we observed that AMPK double knockout cells expressing ARMC10 S45E mutant have more fragmented mitochondria than those in control AMPK double knockout cells. However, this level is still lower than that in WT cells. As suggested by this reviewer, one possible explanation is that ARMC10 S45E is less penetrant in the absence of AMPK directed phosphorylation of Mff. The other non-exclusive explanation is that ARMC10 S45E mutant may only partially restore the function of ARMC10. We added this discussion in the Result section in the revised manuscript.

Reviewer #3 (Remarks to the Author):

I would like to thank the authors for taking the time to carefully look at my comments, I feel the quality of the manuscript has improved due to this effort. Many of my comments have been addressed, however, I feel the BioID workflow is still not described. I urge the authors to provide details on the experimental conditions as well as data analysis, as is customary in the proteomics field and should be requested by the journal. Please find my final comments below.

For the BioID experiments, the protein used as negative control bait should be listed and the parameters used for protein identification as well as the scoring cutoffs employed need to be specified. Also, it is confusing that three baits were described in the experiment and the data of only two was shown in Fig. 6A. Please clarify whether the same sample preparation and mass spectrometric methods were employed here as described in the MS methods section.

Thank you! We used the protein SLX4IP as a nonrelated bait protein in the control group. We added the information about the protein used as negative control in both method and figure legend. In the revised manuscript, we also described in more details about BioID2 experiments in the Method section, including sample preparation, MS running and parameters for protein identification.

In addition, it is still unclear how peptides carrying more than one phosphorylation site were treated and although the source of your GO-terms is now provided, there is still no information available on their fold enrichment, the background proteome used or any significance testing. Hence, the selection of the presented GO terms is still unclear.

Thank you for the comment. The peptides with two or more phosphorylation sites were treated as two or more different phosphopeptides. We added the text in Supplementary Table 1 in the revised manuscript.

Fig. 2c and Supplementary Table 3 presented the function annotation for each protein. As suggested, we added another function assay as shown in Supplementary Figs 3c-3e. It shows the top enriched GO terms and their enrichment p-value with significance conducted using Ingenuity Pathway Analysis.

REVIEWERS' COMMENTS:

Reviewer #2 (Remarks to the Author):

Thank you for addressing these concerns.

Reviewer #3 (Remarks to the Author):

The authors have now addressed all remaining concerns